genomics/bioinformatics

genome reduced-representation, sequence capture enrichment, *Calanus* spp.

**Author for correspondence:**
Marvin Choquet
e-mail: marvin.choquet@nord.no

†Shared first authorship.

# Towards population genomics in non-model species with large genomes: a case study of the marine zooplankton *Calanus finmarchicus*

Marvin Choquet[1,†], Irina Smolina[1,†], Anusha K. S. Dhanasiri[1], Leocadio Blanco-Bercial[2], Martina Kopp[1], Alexander Jueterbock[1], Arvind Y. M. Sundaram[3] and Galice Hoarau[1]

[1]Faculty of Biosciences and Aquaculture, Nord University, Bodø, Norway
[2]Bermuda Institute of Ocean Sciences, St George's, Bermuda
[3]Norwegian Sequencing Centre, Department of Medical Genetics, Oslo University Hospital, Oslo, Norway

MC, 0000-0001-6719-2332; IS, 0000-0002-0205-7663; LB-B, 0000-0003-0658-7183; AJ, 0000-0002-0659-3172

Advances in next-generation sequencing technologies and the development of genome-reduced representation protocols have opened the way to genome-wide population studies in non-model species. However, species with large genomes remain challenging, hampering the development of genomic resources for a number of taxa including marine arthropods. Here, we developed a genome-reduced representation method for the ecologically important marine copepod *Calanus finmarchicus* (haploid genome size of 6.34 Gbp). We optimized a capture enrichment-based protocol based on 2656 single-copy genes, yielding a total of 154 087 high-quality SNPs in *C. finmarchicus* including 62 372 in common among the three locations tested. The set of capture probes was also successfully applied to the congeneric *C. glacialis*. Preliminary analyses of these markers revealed similar levels of genetic diversity between the two *Calanus* species, while populations of *C. glacialis* showed stronger genetic structure compared to *C. finmarchicus*. Using this powerful set of markers, we did not detect any evidence of hybridization between *C. finmarchicus* and *C. glacialis*. Finally, we propose a shortened version of our protocol, offering a promising solution for population genomics studies in non-model species with large genomes.

# 1. Background

Assessment of population genetic metrics for non-model species, and in particular marine zooplankton, has usually been limited to a small number of loci (mostly mitochondrial DNA) [1,2] that may not reflect genome-wide diversity and differentiation [3]. Recent technological advances in next generation sequencing (NGS) have dramatically increased sequencing throughput, reduced associated costs, and together with the development of bioinformatics tools, have opened the door for population genomics studies in any species [4]. Nevertheless, whole-genome sequencing for many individuals of species with genomes greater than 1 Gb remains hampered by cost and bioinformatics challenges associated with the volume of data generated [4,5]. However, as many biological questions can be answered with only a fraction of the genome, genome reduction sequencing methods have become increasingly popular. These methods include amplicon, transcriptome, restriction digest, and capture enrichment sequencing [6–8]. Such methods, not only allow the analysis of 1000s of single nucleotide polymorphism (SNPs) in many individuals [6], but also usually result in higher coverage per locus, and increased accuracy of polymorphism detection compared to whole-genome sequencing approach [9].

Restriction site-associated DNA sequencing protocols (e.g. RAD-seq, [10]; ddRAD-seq, [11]; 2b-RAD, [12]) appear to be suitable for non-model species, as they allow low-cost genotyping of SNPs throughout the genome without allele-specific expression bias in contrast to RNA-seq, and do not require existing genomic resources nor species-specific reagents [4,6,7]. RAD-seq protocols involve an enzymatic digestion of the DNA followed by the selective sequencing of the fragments flanked by restriction enzymes' recognition sites [10]. The double digest RAD-seq uses a double enzymatic digestion of DNA and allows to adjust the number of fragments to be sequenced via the choice of restriction enzymes and the size selection of digested fragments [11]. Although this method presents several advantages, especially when dealing with species with large genomes, the initial requirements in terms of DNA amount and quality may represent a limiting factor for some organisms, such as small planktonic organisms.

Alternatively, sequence capture enrichment, also called targeted resequencing, is a genome-reduced representation protocol that requires only a small amount of DNA for library preparation [13], a great advantage when working with tiny organisms. Different strategies of capture have been developed and are reviewed in Mamanova *et al*. [14]. Overall, the method consists of capturing specific fragments of the genome by hybridization with probes that contain complementary sequences of the targeted sequences [15,16], followed by NGS. Prior knowledge of the sequences targeted is therefore required in order to design the corresponding capture probe set [8,17]. As this can represent a real challenge in the case of non-model species, alternative strategies have been developed, such as using a transcriptome as reference because it is usually easier to assemble than a genome [18] and particularly in the case of species with large genomes. The capture enrichment method offers valuable advantages such as the possibility to use a set of capture probes developed for one species on closely related species with satisfying performance [19–23]. Capture enrichment approaches have also proven effective on historical and degraded DNA [24–27]. Several studies reported high quality of resulting data, consistent loci coverage and, subsequently, accurate SNP calling, when using a capture enrichment-based protocol for reduced genome representation [15,28–30].

In the present study, we developed a genome-reduced representation protocol to pave the way for population genomics studies in the marine copepod *Calanus finmarchicus*. This species dominates the mesozooplankton assemblage of the North Atlantic Ocean in terms of biomass [31] and plays an important role in linking lower and higher trophic levels [32]. Despite *C. finmarchicus* paramount ecological importance, genome-wide studies of the species have been hampered by its large genome (6.34 Gbp haploid; [33]). Its population genetic structure and connectivity have been long-standing subjects of research, reflecting the history of genetic marker development from allozymes [34] and mitochondrial genes [35,36] to microsatellites [37] and a few nuclear SNPs [38]. All studies have suggested high levels of polymorphism and gene flow. However, conclusions have ranged from lack of population genetic structure using six microsatellite loci [37] to a large-scale structure based on 24 SNPs in three nuclear genes [38]. The question of whether there are genetically differentiated populations of *C. finmarchicus* across the North Atlantic Ocean thus remains open and requires a genome-wide approach.

We first applied a ddRAD-seq protocol on pooled *Calanus* individuals from different locations. This protocol requires a high amount and high quality of DNA to start with, but as the amount of DNA extracted from one individual of *Calanus finmarchicus* is rather low, due to the body-size of the organism (typically between 2 and 3 mm), pooling several individuals together was the only option.

**Table 1.** *Calanus finmarchicus* and *C. glacialis* sample information.

| location | method | species | n | collection date | lat. | long. |
|---|---|---|---|---|---|---|
| Barents Sea | Transcriptomic capture | *C. finmarchicus* | 1 | 6 Aug 2012 | 70.50° N | 19.99° E |
| Isfjord (Is) | Genomic capture | *C. finmarchicus* | 8 | 5 June 2016 | 78.32° N | 15.15° E |
| | | *C. glacialis* | 3 | | | |
| Skjerstadfjord (Skj) | Genomic capture | *C. finmarchicus* | 8 | 26 Feb 2016 | 60.72° N | 5.10° E |
| | | *C. glacialis* | 6 | | | |
| Lurefjord (Lure) | Genomic capture | *C. finmarchicus* | 8 | 22 June 2016 | 67.18° N | 15.43° E |
| | | *C. glacialis* | 3 | | | |

The enzyme pair to be used for the digestion was selected based on the results from an *in silico* digestion of the very small fraction of the genome sequenced so far (less than 0.5%). Although we would normally expect a small portion of genome to be sufficient for *in silico* digestion, it seems obvious that the large (and probably duplicated) genome of *C. finmarchicus* may have altered the success of this approach in selecting an optimal restriction enzyme pair. Indeed, in the SimRAD-based method [39], the correspondence between *in silico* and actual digested fragments was not evaluated for cases of large duplicated genomes. Thus, the actual digestion of *C. finmarchicus*'s DNA pools resulted in a very high number of fragments, requiring a costly sequencing effort in order to achieve sufficient coverage for all of them. Therefore, we considered the results of this pilot study not promising enough given the limitations and decided to attempt a different approach. The protocols and results associated with our ddRAD-seq pilot study are available as electronic supplementary material of this paper (supplementary material 1).

Next, we decided to focus on a sequence capture enrichment protocol, and we also tested for cross-species capture hybridization on the closely related *C. glacialis*. The present paper describes the corresponding results. Based on our experience, we propose a simplified method to obtain an informative SNP panel for population genomic studies in non-model species with large genomes.

# 2. Material and methods

## 2.1. Samples and DNA extraction

Zooplankton samples were collected from four locations (table 1) by vertical tows between 0 and 200 m depth using WP2 [40] or similar nets with mesh size of 200 μm. Samples were immediately preserved in 95% undenatured ethanol, with subsequent change of ethanol after 24 h. Genomic DNA was extracted individually using the E.Z.N.A. Insect DNA Kit (Omega Bio-Tek) according to the manufacturer's instructions. *Calanus* species identification was performed for each individual using a set of six nuclear insertion–deletion markers (InDels) in a multiplex PCR following the protocol described in Smolina *et al.* [41].

## 2.2. Development of a genomic reference for *Calanus finmarchicus*

### 2.2.1. Probe set design for transcriptome-based sequence capture

So far, no good quality genomic reference is available for *C. finmarchicus*, but three transcriptomes have been published [42–44]. We used the transcriptome from Lenz *et al.* [43], which is the most complete currently available, to design a set of probes to capture, sequence and assemble genes of interest into a custom genomic reference. From the transcriptome, we selected all sequences ≥750 bp long (=29 518 sequences), to which we added the 38 unique transcripts known to be involved in thermal stress response of *C. finmarchicus* [42]. We blasted (blastn in Geneious v. 9.1.8) each of these transcripts against the whole transcriptome and kept only unique sequences in order to reduce false-positive SNPs from paralogous and repeated regions. Then, we trimmed the resulting 18 588 sequences to the first 200 bp, to target the 5′UTR regions, supposedly enriched in SNPs [45]. Our design of 3 717 600 bp in total was then sent to Roche NimbleGen Inc. (Madison, WI) to produce 120-mer sequence-capture probes.

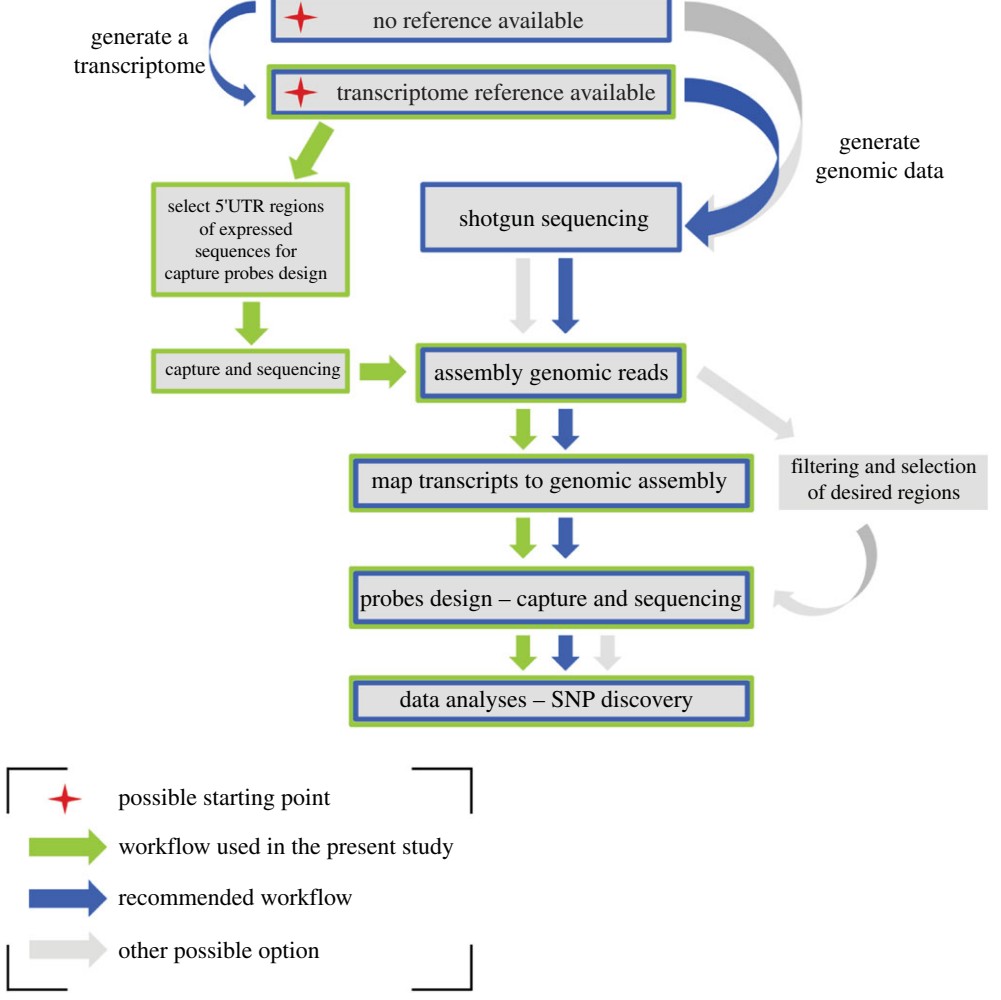

**Figure 1.** Alternative capture-enrichment based workflows for SNP discovery in non-model species with large genomes.

### 2.2.2. Library preparation and sequence capture

A library was prepared from a single individual of *C. finmarchicus* from the Barents Sea (table 1), according to the manufacturer's protocol (*NimbleGen SeqCap EZ Library SR version 4.2*) (see details of library preparation and capture in supplementary material 2). The captured DNA was sequenced on a MiSeq sequencer (Illumina) with the $2 \times 300$ bp v. 3 chemistry.

### 2.2.3. Evaluation of the capture efficiency

We mapped the 15 556 070 raw reads obtained from the sequencing to the 29 556 full-length transcriptomic contigs initially used for the capture design using the BWA-MEM (v. 0.7.16) tool in default mode [46]. The fact that only 30.12% of the reads mapped uniquely with high-quality score strengthens the need for a genomic-based reference. Therefore, raw reads were filtered to remove duplicates and low complexity sequences using PRINSEQ (v. 0.20.4) [47] and then assembled using the MaSuRCA assembler (v. 3.2.2) [48] to be used as a reference for a second probe design (figure 1).

## 2.3. Genome-based sequence capture

### 2.3.1. Probe set design

From the genomic data generated by the previous sequencing, we identified all the transcripts successfully captured and sequenced. To achieve this, we first mapped all the transcriptomic reads available for *C. finmarchicus* on NCBI (https://www.ncbi.nlm.nih.gov/; Ref: PRJNA236528) to the 29 556 full-length transcriptomic sequences using Bowtie2 (version 2.2.3) [49]. Then, to identify

targeted genes that were successfully captured and sequenced, the 33 294 898 RNAseq reads that mapped to the selected transcripts were mapped to our MaSuRCA assembly of genomic data using TopHat RNAseq splice-aware mapper (version 2.1.1) [50]. This resulted in 9 225 593 reads that were mapped to 36 223 contigs. These 36 223 contigs were then blasted (blastn in Geneious v9.1.8) against themselves in order to keep only single-copy genes, resulting in 3500 contigs with only 1 hit (self-hit). We performed the second blast of these 3500 contigs against the full MaSuRCA assembly (generated in previous section), and we selected the 2223 contigs with 1 hit and 433 other contigs with more than one hit but having 97% or more pairwise identity. We finally obtained a total of 2656 contigs with length from 302 to 3029 bp that we trimmed to a maximum length of 1500 bp. The final design of 2656 sequences, representing 2 106 591 bp, was then sent to the MYcroarray MYbaits company (Inc., MI, USA) to produce 80-mer sequence-capture probes.

### 2.3.2. Library preparation

The second run of capture was performed on a total of 36 individual libraries, including 24 *C. finmarchicus* individuals from three locations, and 12 *C. glacialis* individuals from the same three locations (table 1). Libraries were prepared using the NEXTflex$^{TM}$ Rapid Pre-Capture Combo Kit (Bioo Scientific, Austin, TX, USA) (see details in supplementary material 2). Individually indexed libraries were then pooled per species, before proceeding to capture. As *C. finmarchicus* has an estimated genome size of 6.34 Gbp (haploid), while *C. glacialis* has an estimated genome size of 11.83 Gbp [33], we reduced the number of libraries of *C. glacialis* to be pooled for the capture reaction to ensure that similar genome copy numbers are present. The sequence capture was performed for each pool/species according to the MYcroarray MYbaits protocol (http://www.mycroarray.com/pdf/MYbaits-manual-v3.pdf) with a few adjustments (detailed in supplementary material 2) to maximize efficiency. Finally, the two pools were mixed together in equal proportions and sequenced on a NextSeq 550 (Illumina) with the $2 \times 150$ bp mid-output kit v. 2.

### 2.3.3. Evaluation of the capture efficiency

The NextSeq sequences were demultiplexed and mapped directly to the MaSuRCA assembly using BWA-MEM (v. 0.7.16) [46]. Only the reads mapping uniquely to the reference, concordantly, and in pairs were kept. Duplicates were removed using Picard tools (http://broadinstitute.github.io/picard), and mapped reads were realigned around InDels using GATK (v. 3.7) [51]. The percentage of high-quality reads mapping back to the reference was more satisfying than previously with 38% of *C. finmarchicus* reads on average mapping uniquely and without duplicates, and 23% for *C. glacialis*.

## 2.4. SNP genotyping and application for population genomics

### 2.4.1. Genomic variation analyses

Variants were called for all individuals of both species together at once using the HaplotypeCaller [52] implemented in GATK (v. 3.7). In order to make accurate estimates of genetic diversity, we forced HaplotypeCaller (GATK) to also output the non-variant sites, together with the variants, in the resulting VCF file. Using VCFtools (v. 0.1.13) [53], we filtered the sites to keep only those with mean depth values (over all individuals) greater than or equal to $5\times$. Among these, sites with more than 20% of missing data were excluded, which means that we kept only the sites represented in at least 80% of the genotypes.

The resulting file was used to estimate nucleotide diversity ($\pi$) for each species and location separately. Nucleotide diversity was estimated on a per-site basis and averaged in 780 bp windows (average of contig size distribution) using only the sites that passed the filtering. We reported the mean of $\pi$ across windows for each population, with VCFtools (v. 0.1.13).

Observed heterozygosities (proportion of heterozygous sites) at variant sites were calculated on a per-SNP level in each individual and averaged over all positions present in both species together, using VCFtools (option –het; v. 0.1.13).

### 2.4.2. Population structure and gene flow analyses

Once more, variants were called for all individuals of both species together at once, using the HaplotypeCaller [52] implemented in GATK (v. 3.7). With GATK and VCFtools (v. 0.1.13) [53], raw

variants were hard-filtered for different quality parameters (see details in supplementary material 3), InDels were removed, variants phased and only SNPs covered between 5× and [average + 2*standard deviation]× were kept. Sites present in less than 80% of genotypes were filtered out. SNPs with minor allele frequency less than 0.05 were removed. The numbers of SNPs present in each species, in each location, and shared by both species and among locations were then calculated with BCFtools (v. 1.6). The command line scripts used for data processing are supplied in supplementary material 3.

The filtered SNPs were pruned based on linkage disequilibrium (LD) in sliding windows of 50 markers, five markers at a time with a $R^2$ threshold of 0.8. This dataset was used to investigate the potential presence of hybrids between *C. finmarchicus* and *C. glacialis* by running ADMIXTURE (v. 1.3.0) [54].

For the next analysis, we re-used the VCF file containing all the filtered SNPs before the pruning, and we split it per species. The two resulting files were then LD-pruned in the same way as in the previous step. The resulting markers were used in two principal component analyses (PCA) (one per species), performed with PLINK (v. 1.9) [55,56].

For calculating the global weighted $F_{ST}$ [56] in each species, only one variant site per contig was randomly selected using a PERL script [57], to avoid giving more weight to contigs with more variants (i.e. probably linked variants). Global weighted $F_{ST}$ was then calculated in PLINK. Distributions of the $F_{ST}$ values were obtained after 1000 iterations of the procedure (therefore different combinations of SNPs from each contig), and median, average and quartiles calculated for each species (supplementary material 3).

### 2.4.3. Test for selection

Candidate SNP loci under selection were identified using BayeScan (v. 2.1) [58–60] for each species separately from the non-LD-pruned SNPs. The software compares allele frequencies among populations to determine which genetic markers are outliers and thus most likely to be under selection. In complement, we used VCFtools (v. 0.1.13) for calculating a site frequency spectrum of all SNPs per locations and species.

# 3. Results

## 3.1. Genome-based capture efficiency

The 36 libraries (table 1) yielded on average 4.3 million reads per individual for *C. finmarchicus* ($N = 24$), and 16.8 million reads for *C. glacialis* ($N = 12$) (table 2). For *C. finmarchicus*, an average of 1.6 million reads mapped uniquely to the reference. This represents on average 38% (32.7% to 43%) of the initial number of reads sequenced per individual (table 2). For *C. glacialis*, 3.8 million reads mapped on average per individual. This represents on average 23% (20.9% to 25.3%) of the initial number of reads sequenced per individual (table 2).

After variant calling and hard-filtering, 154 087 SNPs with sufficient coverage were identified for *C. finmarchicus*, ranging from 95 453 to 108 131 SNPs per location (table 3) and distributed across 4603 contigs (supplementary material 2: electronic supplementary material, figure S3). A total of 62 372 SNPs were in common among all three locations (table 3). For *C. glacialis*, 121 872 SNPs passed the hard-filtering steps and were sufficiently covered, ranging from 91 923 to 107 752 SNPs per location (table 3). These SNPs were distributed across 5363 contigs (supplementary material 2: electronic supplementary material, figure S3). A total of 80 319 SNPs were in common among all three locations (table 3). Furthermore, 60 452 SNPs were shared between *C. finmarchicus* and *C. glacialis* (table 3).

## 3.2. Population genomics results

### 3.2.1. Genomic variation

After filtering steps, nucleotide diversity estimates were calculated from a total of 316 019 sites (variants and non-variants), for each population in each species. The index $\pi$ revealed similar levels of genetic diversity between species and among locations (figure 2).

A total of 118 196 variant sites were used for calculating the mean individual observed heterozygosities. The obtained averages were very similar between species and among locations (figure 3), ranging from 0.089 to 0.16 for *C. finmarchicus* and from 0.1 to 0.147 for *C. glacialis*.

**Table 2.** Efficiency of the transcriptome-based and genome-based capture enrichment for *Calanus finmarchicus* and *C. glacialis*. Raw reads: total number of sequenced reads used for mapping; % HQ-mapped reads: reads that mapped to a unique site in the genome reference and in proper pairs without duplicates; on-target rate: proportion of reads on target within HQ-mapped reads; global % reads on target: proportion of reads on target extrapolated to the total number of reads sequenced; mean depth of coverage on target: mean depth of coverage of targeted contigs.

| individual | raw reads | % HQ-mapped reads | on-target rate | global % reads on target | mean depth of coverage on target | NCBI BioSample accessions |
|---|---|---|---|---|---|---|
| *C. finmarchicus*—transcriptomic capture | | | | | | |
| CfinPC13_pop1 | 15 556 070 | 30.12% | 83.81% | 25.24% | | SAMN08924867 |
| *C. finmarchicus*—genomic capture | | | | | | |
| CF_Is_1 | 4 181 938 | 39.06% | 79.17% | 30.93% | 83.36 | SAMN08924868 |
| CF_Is_2 | 4 219 268 | 40.54% | 79.53% | 32.24% | 117.65 | SAMN08924869 |
| CF_Is_3 | 3 667 228 | 38.55% | 82.53% | 31.82% | 87.71 | SAMN08924870 |
| CF_Is_4 | 5 119 056 | 40.44% | 79.98% | 32.34% | 106.95 | SAMN08924871 |
| CF_Is_5 | 5 872 096 | 40.62% | 77.46% | 31.46% | 117.65 | SAMN08924872 |
| CF_Is_6 | 5 184 258 | 40.28% | 80.36% | 32.37% | 107.92 | SAMN08924873 |
| CF_Is_7 | 4 678 720 | 43.04% | 73.70% | 31.72% | 93.8 | SAMN08924874 |
| CF_Is_8 | 2 702 248 | 41.00% | 76.36% | 31.31% | 54.04 | SAMN08924875 |
| CF_Lure_17 | 2 093 340 | 38.17% | 70.69% | 26.98% | 35.43 | SAMN08924876 |
| CF_Lure_18 | 1 329 222 | 35.97% | 78.53% | 28.25% | 24.16 | SAMN08924877 |
| CF_Lure_19 | 3 563 372 | 36.63% | 79.45% | 29.10% | 66.81 | SAMN08924878 |
| CF_Lure_20 | 2 395 550 | 33.47% | 76.04% | 25.45% | 38.74 | SAMN08924879 |
| CF_Lure_21 | 3 031 526 | 32.69% | 74.53% | 24.36% | 47.57 | SAMN08924880 |
| CF_Lure_22 | 2 800 918 | 33.67% | 72.88% | 24.54% | 44.07 | SAMN08924881 |
| CF_Lure_23 | 1 267 786 | 38.08% | 76.00% | 28.94% | 23.55 | SAMN08924882 |
| CF_Lure_24 | 3 518 314 | 40.29% | 74.37% | 29.96% | 67.84 | SAMN08924883 |
| CF_Skj_33 | 3 741 466 | 36.32% | 69.88% | 25.38% | 60.14 | SAMN08924884 |
| CF_Skj_34 | 3 438 886 | 39.34% | 72.75% | 28.62% | 62.89 | SAMN08924885 |
| CF_Skj_35 | 3 028 598 | 35.66% | 75.73% | 27.01% | 52.10 | SAMN08924886 |
| CF_Skj_36 | 9 028 836 | 35.55% | 71.53% | 25.43% | 145.25 | SAMN08924887 |
| CF_Skj_37 | 8 244 400 | 34.43% | 72.07% | 24.82% | 131.03 | SAMN08924888 |
| CF_Skj_38 | 6 805 150 | 33.96% | 73.29% | 24.89% | 108.55 | SAMN08924889 |
| CF_Skj_39 | 6 287 262 | 35.64% | 66.94% | 23.86% | 92.97 | SAMN08924890 |
| CF_Skj_40 | 7 023 836 | 39.19% | 62.85% | 24.63% | 106 | SAMN08924891 |
| *average* | *4 300 970* | *38%* | *75%* | *28%* | *78.17* | |
| *C. glacialis*—genomic capture | | | | | | |
| CG_Is_10 | 13 819 538 | 20.95% | 56.00% | 11.73% | 96.82 | SAMN08924892 |
| CG_Is_11 | 7 741 988 | 25.34% | 62.48% | 15.83% | 74.29 | SAMN08924893 |
| CG_Is_16 | 5 230 852 | 24.20% | 64.09% | 15.51% | 49.31 | SAMN08924894 |
| CG_Lure_28 | 5 132 518 | 25.02% | 61.66% | 15.43% | 47.59 | SAMN08924895 |
| CG_Lure_29 | 27 796 636 | 23.88% | 54.63% | 13.04% | 215.97 | SAMN08924896 |
| CG_Lure_32 | 20 645 638 | 22.50% | 57.83% | 13.01% | 160.93 | SAMN08924897 |
| CG_Skj_43 | 18 412 870 | 21.08% | 51.11% | 10.77% | 115.29 | SAMN08924898 |
| CG_Skj_44 | 20 791 734 | 22.84% | 53.91% | 12.32% | 150.80 | SAMN08924899 |

(*Continued.*)

**Table 2.** (Continued.)

| individual | raw reads | % HQ-mapped reads | on-target rate | global % reads on target | mean depth of coverage on target | NCBI BioSample accessions |
|---|---|---|---|---|---|---|
| CG_Skj_45 | 20 389 800 | 23.18% | 58.13% | 13.48% | 164.03 | SAMN08924900 |
| CG_Skj_46 | 18 812 850 | 22.14% | 53.55% | 11.86% | 131.58 | SAMN08924901 |
| CG_Skj_47 | 19 203 884 | 21.96% | 55.19% | 12.12% | 137.95 | SAMN08924902 |
| CG_Skj_48 | 23 482 634 | 22.98% | 49.16% | 11.30% | 153.75 | SAMN08924903 |
| *average* | *16 788 412* | *23%* | *56%* | *13%* | *124.86* | |

**Table 3.** Summary of discovered SNPs using genome-based capture enrichment after hard-filtering, phasing and coverage filtering.

| | species | | | |
|---|---|---|---|---|
| | *C. finmarchicus* | | *C. glacialis* | |
| | *n* indiv. | total # SNPs | *n* indiv. | total # SNPs |
| location | | | | |
| Isfjord | 8 | 104 346 | 3 | 91 923 |
| Skjerstadfjord | 8 | 108 131 | 6 | 107 752 |
| Lurefjord | 8 | 95 453 | 3 | 98 331 |
| SNPs per species | | 154 087 | | 121 872 |
| SNPs in common among three locations | | 62 372 | | 80 319 |
| SNPs in common between species | | | 60 452 | |

### 3.2.2. Population structure and gene flow

The ADMIXTURE analysis, based on 37 710 SNPs shows a very clear clustering per species, without apparent gene flow (figure 4).

The PCA performed for *C. finmarchicus*, based on 34 449 SNPs, shows no noticeable differentiation among individuals from different locations. Two outliers were identified as individuals from Lurefjord (CF_Lure_18 and CF_Lure_23) (figure 5a). The PCA performed for *C. glacialis*, based on 17 035 SNPs, shows the differentiation of two groups of individuals, corresponding to the locations of Isfjord and Skjerstadfjord. Individuals from Isfjord are differentiated from the two other locations on the PC1 (11.91%) (figure 5b).

Estimation of genetic differentiation (weighted $F_{ST}$) for each species among the same three locations was much higher (about six times higher) for *C. glacialis* (mean = median, $F_{ST} = 0.019$; 4113 SNPs per iteration) compared to *C. finmarchicus* (mean = median, $F_{ST} = 0.003$; 4216 SNPs per iteration), and statistically significant in both species ($p < 0.001$) (figure 6).

### 3.2.3. Selection

Test for SNP loci under selection using BayeScan revealed no loci under recent and strong positive selection in *C. finmarchicus* out of 46 544 SNPs analysed (figure 7a). In *C. glacialis*, three loci out of 49 742 (0.006%) are likely to be under recent and strong positive selection (figure 7b).

The site frequency spectrum revealed no apparent selection in either species (supplementary material 2, electronic supplementary material, figures S4 and S5); however, the low number of individuals should be taken into account when drawing conclusions from the site frequency spectrum diagrams.

## 4. Discussion

Zooplankton organisms represent a key link in marine food webs and play a crucial role in marine ecosystems. They are often used as beacons of climate changes, therefore understanding their

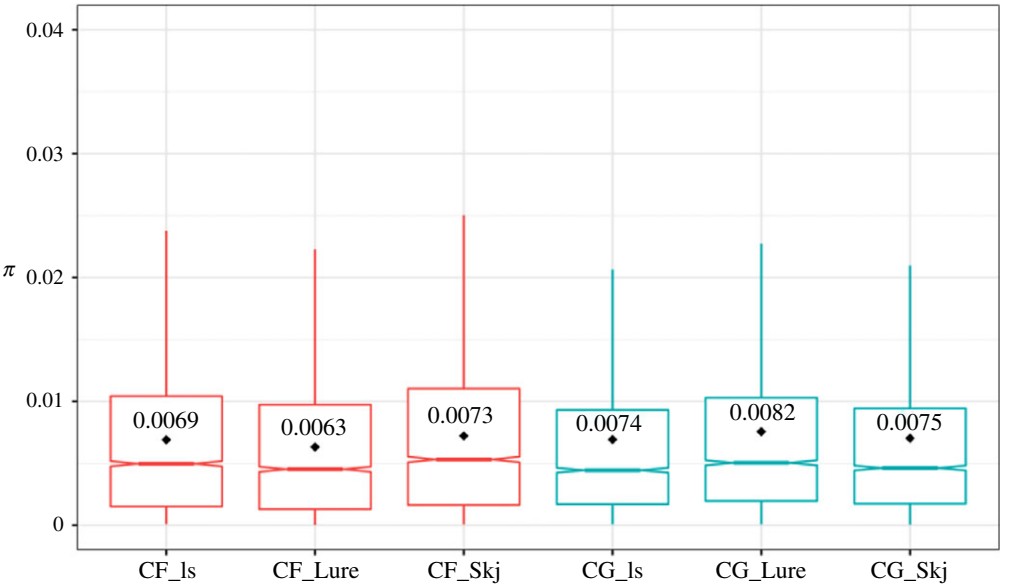

**Figure 2.** Nucleotide diversity ($\pi$) in each population of *Calanus finmarchicus* (red) and *C. glacialis* (blue) estimated from 780 bp non-overlapping windows of variant and non-variant sites. Each box plot notch represents the median. Mean values per location are written in each box.

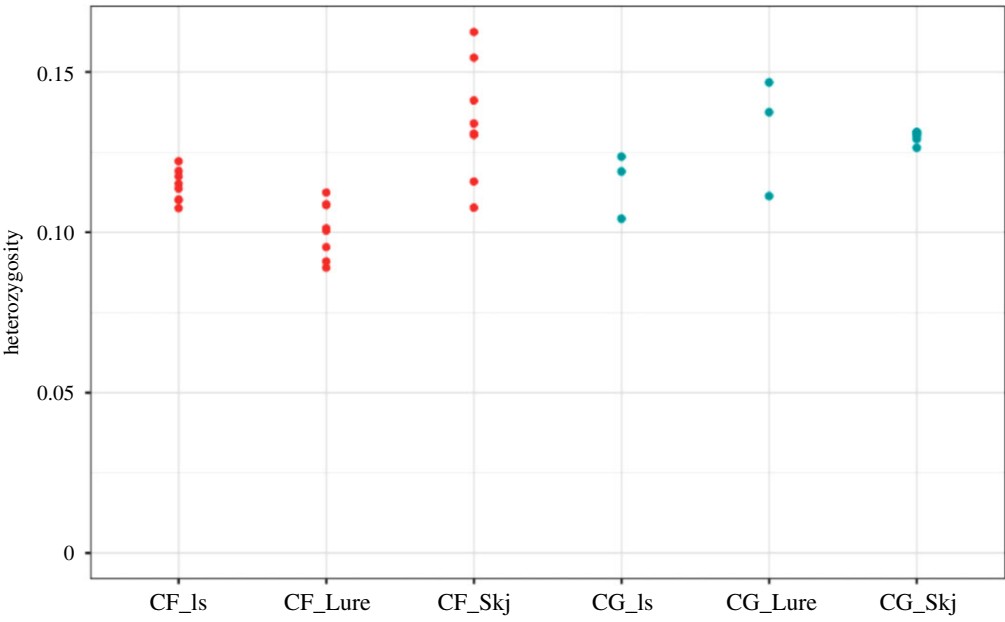

**Figure 3.** Individual heterozygosity levels within *Calanus finmarchicus* (red) and *C. glacialis* locations (blue). Mean proportion of heterozygous sites observed per individual.

population structure and genetic connectivity is critical. However, this task may be challenging, as gene flow can be high in zooplankton species and often results in subtle patterns of genetic structure not necessarily detectable with only a few markers [61,62], thus requiring a genomic approach [63]. So far, technical difficulties linked to the large genome sizes of many of these organisms, particularly in the Arthropoda phylum, have hampered population genomics studies (reviewed by [64]). In the present study, our aim was to identify an efficient genome reduction method to obtain a sufficiently large number of SNPs to conduct robust population structure studies on *Calanus finmarchicus*.

Our results suggest that a sequence capture protocol may be the easiest and most effective way to deal with non-model species with large genomes, especially when it comes to small-sized organisms. Indeed, our optimized protocol yielded more than 154 000 SNP markers for the targeted species. This number represents on average seven times more high-quality SNPs than what we obtained with our ddRAD-seq tentative approach for a comparable sequencing effort (supplementary material 1).

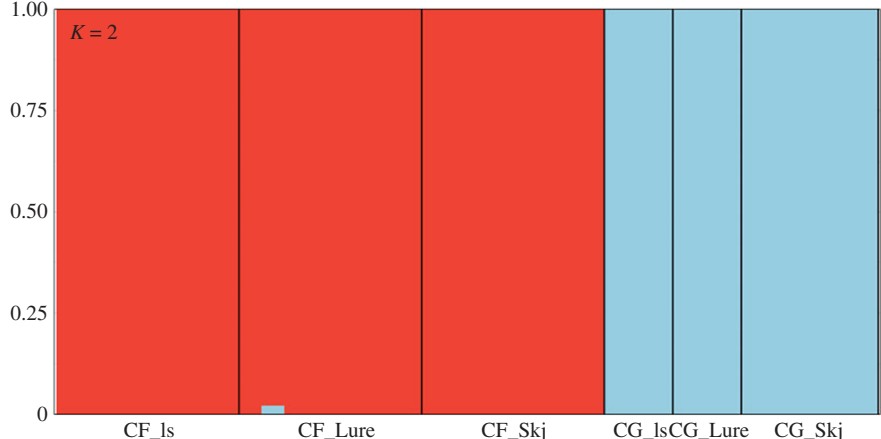

**Figure 4.** ADMIXTURE analysis of SNP markers from co-occurring *Calanus finmarchicus* and *C. glacialis* individuals from the same three geographical locations (*K* = 2). The analysis was performed using a total of 37 710 SNPs. Each group of individuals from the same geographical location are represented by a vertical bar, in red for *C. finmarchicus* and in blue for *C. glacialis*. For *C. finmarchicus*, there are eight individuals per location. For *C. glacialis*, there are three individuals for the locations CG_ls (Isfjord) and CG_Lure (Lurefjord) and six individuals for CG_Skj (Skjerstadfjord). This plot shows two distinct clusters, in two different colours, corresponding to the two different species. This clear distinction proves there is no hybrid in the dataset.

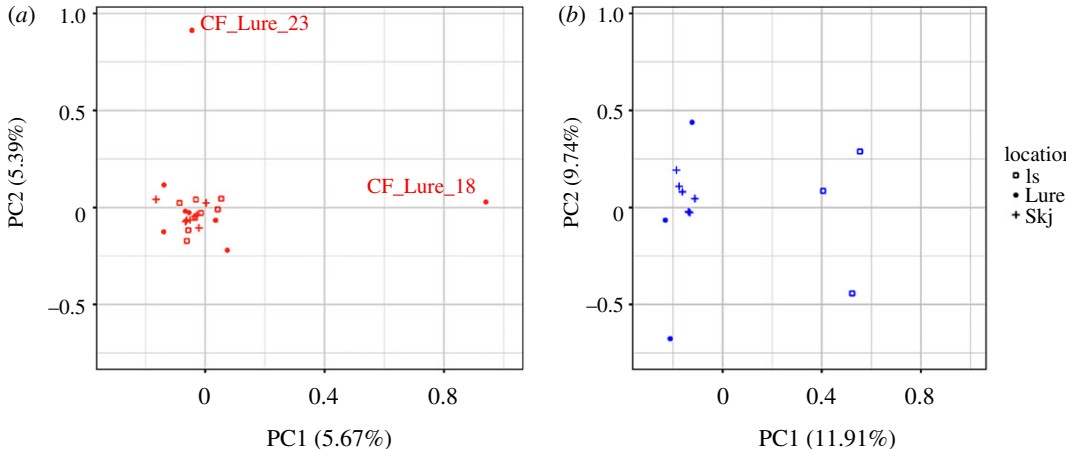

**Figure 5.** Principal component analyses (plot of 2 first components) performed with SNP markers from *Calanus finmarchicus* (*a*) and *C. glacialis* (*b*) individuals from three locations. The 24 individuals of *C. finmarchicus* are displayed in red colour while the 12 *C. glacialis* individuals are displayed in blue colour. Each shape represents a distinct location.

Furthermore, the capture-based protocol yielded 70 times more contigs bearing SNPs (on average), thus resulting in a higher number of unlinked loci. One of the main challenges with the RAD-seq method was the DNA requirement, forcing us to pool individuals due to the limited amount of DNA available per individual. This is clearly an advantage of capture enrichment protocols [13], as a very small amount (less than 10 ng) or even partially degraded DNA can be used [27]. Sequence capture was also very successful for the congeneric species *C. glacialis*, with *ca* 122 000 SNPs identified. Besides, the physical proximity of many of the SNPs identified with sequence capture (4603 contigs for *C. finmarchicus*; 5363 contigs for *C. glacialis*) opens up the possibility to infer the precise sequence (phase) of alleles on each homologous copy of a chromosome [65,66]. Such phased haplotype can then be used to infer ancestry and demographic history [67] or to detect selection [68].

Although transcriptome-based capture sequencing can be successful (e.g. [18,69]), it typically requires a reference genome of a closely related species to identify intron-exon boundaries. Absence of such genomic information for most of zooplankton species (reviewed in Bucklin *et al*. [64]) and limited success of transcriptome-based capture of *Calanus* exemplified in the present study, suggest that the two-step capture protocol we used, offers a good compromise (figure 1). Moreover, with the constant reduction of sequencing costs, this method can be further simplified by generating genomic reference data directly by shotgun sequencing and aligning genomic and transcriptomic sequences in order to

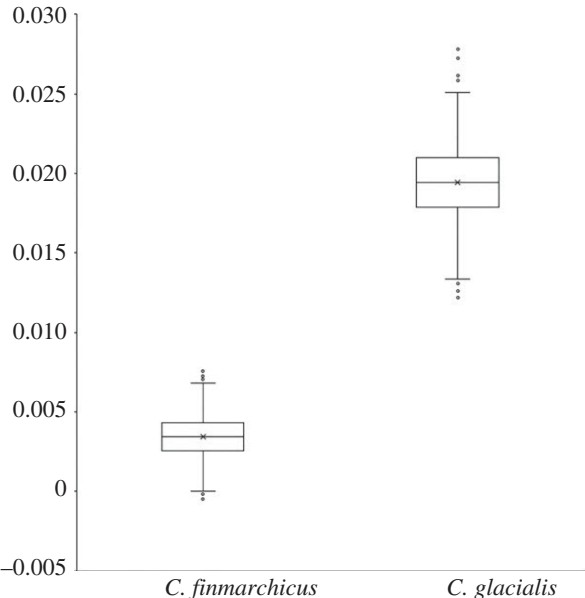

**Figure 6.** Distribution of the global weighted index of genetic differentiation $F_{ST}$ within *Calanus finmarchicus* and *C. glacialis*. The distribution of the global weighted $F_{ST}$ within each species was calculated after 1000 iterations, selecting one random SNP per contig for all contigs for each iteration. Boxes indicate the first, second (median) and third quartiles, with the average $F_{ST}$ indicated by the 'x'; whiskers show 1.5 times the interquartile range above and below the third and first quartile respectively. Data above or below the whiskers range were considered outliers, indicated as circles. Only two iterations marginally reached values less than 0 for *C. finmarchicus*.

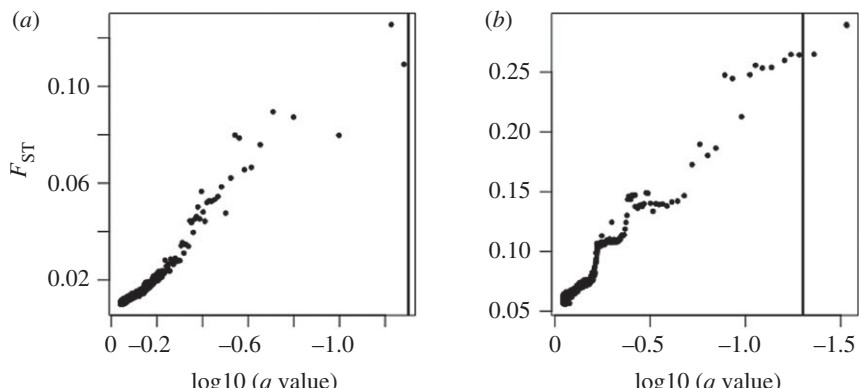

**Figure 7.** Identification of SNPs under recent and/or strong positive selection with BayeScan in *Calanus finmarchicus* (*a*) and *C. glacialis* (*b*). Each locus's $F_{ST}$ value is plotted against the log10 of the corresponding *q*-value (false discovery rate (FDR), analogue of the *p*-value). The vertical bar indicates the threshold for FDR = 0.05 value used to identify outlier SNPs, represented on the right side of the bar.

target mainly genic or anonymous intergenic regions, depending on the purpose of the study. The corresponding shortened workflow, illustrated in figure 1, has been tested on another zooplankton species, the pteropod *Limacina bulimoides*, and preliminary results are promising [70].

The vast majority (greater than 99.99%) of the SNPs identified with the capture-based protocol did not show any sign of recent and/or strong positive selection. Assessment of genetic diversity and heterozygosity levels revealed very similar results between the two species of *Calanus* (figures 2 and 3). Although the levels of genomic variation are comparable between the two species, the PCAs show contrasting preliminary patterns of genetic structure within the two species. Indeed, there is higher inter-individual variation in *C. glacialis* and also higher inter-location differentiation than for *C. finmarchicus*. Individuals of *C. finmarchicus* appear genetically close to one another independently of their geographical origin, except for two outliers, CF_Lure_18 and CF_Lure_23, both from the Lurefjord location. Their position in the PCA can easily be explained by the relative lack of usable data for these two individuals, as they have the lowest numbers of raw reads and lowest values of

mean depth of coverage among all individuals sequenced (table 2). Samples from the Lurefjord (southern Norway) were collected in June, when temperatures were high, and samples may have suffered from the summer conditions before they could be appropriately stored at cold temperature. This could have led to some degradation of copepods' DNA resulting in lower success of sequencing. In contrast to *C. finmarchicus*, gene flow among *C. glacialis* locations seems more limited. In particular, and interestingly, the Isfjord (Svalbard) appears genetically well differentiated from the two other locations on the first axis of the PCA (PC1: approx. 12%). Individuals from the Skjerstadfjord are clustered closely together, while the individuals from the Lurefjord are more distanced from one another. Both $F_{ST}$ and PCAs suggest a more recent and stronger genetic structure for *C. glacialis* compared to *C. finmarchicus* populations. It is important to keep in mind though, that we have been sampling *C. glacialis* genome using *C. finmarchicus* originated probes. Consequently, we may have missed some naturally more variable regions in *C. glacialis* by capturing mostly regions conserved enough between species to be recognized by the capture probes; another possibility is that due to the lower number of individuals we would be missing more variants (especially those less frequent) in *C. glacialis* (see fig. 2 in [71]). These are cases of ascertainment bias [71,72]. Only the investigation of more individuals from the entire distribution range of the species will help to evaluate the significance of this effect. Nonetheless, the obtained results are in line with microsatellite data validating the usefulness of the SNPs. Indeed, Choquet *et al.* [73] reported a global $F_{ST}$ 7.5 times higher for *C. glacialis* populations compared to *C. finmarchicus* for the same three locations. The SNPs dataset shows a 6× difference between the two species, but higher precision is expected given the number of markers (4000 SNPs versus six microsatellite loci) [63]. However, the present study focused at developing a suitable method for investigating genetic connectivity in a non-model species with a large genome and is limited by the sampling size. A larger sampling scale is required to understand the population structure of the different *Calanus* species.

Population genomics studies of marine zooplankton have been very scarce [64]. In the copepod *Centropages typicus*, a 2b-RAD-seq approach yielding 675 SNPs revealed genetic structure between the northwest and the northeast Atlantic Ocean [61], which was in contrast with results from a previous study based on COI and ITS markers [74]. Another study used RAD-seq on the Antarctic krill *Euphausia superba* [75], and found no population structure across the whole Southern Ocean. However, the authors reported on the many challenges they went through by using RAD-seq on a very large and complex genome (*ca* 24 Gbp haploid) with no primary reference available, particularly due to the fact that most of the markers they discovered were from multicopy genomic regions and had to be removed from downstream analyses [75].

Finally, the obtained SNP set that is shared by *C. finmarchicus* and *C. glacialis* represents a very powerful tool to investigate the potential for hybridization and introgression between the two species. Indeed, using microsatellites, the presence of hybrids between *C. finmarchicus* and *C. glacialis* has been suggested at the Canadian east coast [76]. However, the genotyping of more than 8000 individuals using six co-dominant nuclear InDel markers developed from both species never detected any hybrids, in any of the 85 locations investigated in the North-Atlantic and Arctic Ocean [73,77]. The change of scale, between a few markers (six InDels or 10 microsatellites), and tens of thousands of markers described here, is considerable, and from the limited dataset obtained in the present study, there is no indication of inter-species hybridization. However, this question needs to be addressed further using samples from the two species' entire distribution ranges, and the presently identified set of genome-wide SNPs will be a powerful instrument in this pursuit.

Ethics. No permissions were required to carry out the fieldwork related to this study. No 'Animal Care Protocol' was required for copepods.

Data accessibility. Raw sequencing data from the ddRAD-seq approach are available on NCBI (https://www.ncbi.nlm.nih.gov/), reference: Bioproject ID PRJNA304215. Raw sequencing data from both transcriptomic and genomic captures are available on NCBI, reference: BioProject ID PRJNA449998, and in the SRA database: SRP139901.

Authors' contributions. I.S. and M.C. contributed equally to the study design, the molecular work, sequencing data analyses and the manuscript writing. G.H. designed the study, contributed to the data analyses and to the manuscript writing, and supervised the whole project. A.K.S.D. and M.K. contributed to the development of the molecular methods and to the molecular work. L.B.-B., A.Y.M.S. and A.J. contributed to bioinformatics analyses. All authors contributed to the manuscript and gave final approval for publication.

Competing interests. The authors declare no competing interests.

Funding. This work was funded by the Norwegian Research Council (HAVKYST 216578), EU-FP7 Eurobasin (Grant agreement 264933) and Nord University (Bodø – Norway).

Acknowledgements. We would like to warmly acknowledge Ann Bucklin (University of Connecticut) for the samples she provided and for her comments on the manuscript. We thank Maja Hatlebakk (The University Centre in Svalbard) and Morten Krogstad (Nord University) for collecting samples used in the capture-enrichment part of this study. We thank Torkel Gissel Nielsen (Danish Technical University) for collection of samples around Greenland, and Ebru Unal (University of Connecticut and Mystic Aquarium) for sorting of samples from the Gulf of St. Lawrence. Special thanks to the captains, crews and scientists who participated in the research cruises providing samples used for this study: RV Johan Hjort cruise 201208 and RV GO Sars cruise 201305. We are grateful to the two anonymous reviewers and to the editor for their constructive comments and suggestions. Particularly, we would like to thank Alexander Nater for his thorough review that largely contributed to the quality of this manuscript.

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
