## [Reviewer comments · Royal Society Open Science]

Review History

RSOS-180608.R0 (Original submission)

Review form: Reviewer 1

Is the manuscript scientifically sound in its present form?

No

Are the interpretations and conclusions justified by the results?

No

Is the language acceptable?

Yes

Is it clear how to access all supporting data?

Yes

Do you have any ethical concerns with this paper?

No

Have you any concerns about statistical analyses in this paper?

No

Recommendation?

Major revision is needed (please make suggestions in comments)

Comments to the Author(s)

Generating genomic resources for the research community to conduct more powerful population genetic studies in non-model organisms is certainly an important task. In this study, the authors present their workflow to generate a panel of genome-wide markers in the marine copepod *Calanus finmarchicus*, which has been hampered so far by the relatively large genome size of this organism. The authors compare the suitability of two commonly used reduced-representation techniques, ddRAD and targeted-capture. Generally, I think this study has merit, but there are currently too many confounding factors for a meaningful comparison. Most of the differences between number of SNPs found with both methods stem probably from a bad/difficult design, together with differences in sequencing effort, mapping and variant calling, and choice of individuals/populations. I would advise the authors to tone down the comparative aspect, and instead address some of the biological questions that have motivated this study in the first place. After all, the authors set out to gauge the suitability of both methods in order to be able to address questions about the population structure/connectivity of this ecologically important organism. Showing some of those results and comparing them to previous studies might help to demonstrate how important a genome-wide perspective is and give this manuscript more impact. In my opinion, given the many caveats, the main message and conclusions about the suitability of one method over the other is not strongly supported and apart from a fairer treatment throughout the manuscript, I would like to see results on the biological questions before I can recommend this manuscript for publication. That being said, I am happy that the decision whether the slight methodological modifications to the "standard" procedures are sufficient to publish this manuscript in its present form as a method/protocol lies with the editor.

Major comments:

1.) Main message and addressing the biological questions

I think it would help if the authors could make it more clear what the main rationale and message of the manuscript is. For example, in the abstract the authors clearly and directly compare the two methods, but in the last paragraph of the introduction (p. 5, ll. 9-11), the authors state "The present paper does not aim to directly compare the two approaches, ...". Moreover, the authors highlight the biological questions (e.g. population connectivity, structure) that their approaches are supposed to help with in the introduction (p.4, ll. 49-51), but no analyses / results for this are shown. Based on the sampling, I would think that the authors could have easily addressed some of the biological aspects (e.g. show a PCA, Admixture plot of all populations and compare it to previous studies), which would help to gauge if the genome-wide approach is really more suited and needed to study population structure in marine arthropods. I note, that there is hardly any doubt that a genome-wide approach is superior, it would just be nice if the authors could show this. At the moment it almost appears as if the authors want to simply double their scientific output by having a purely methodological paper now, while saving the results for another manuscript. I would strongly advise against that, unless it can be justified why this is done (e.g. more data is collected at the moment from more populations, whereas this is just a pilot study to see if the method works, etc.).

2.) Comparison of the two approaches and confounding factors

The authors make it sound as if ddRADseq is generally worse than the capture-based approach. For example, already in the introduction the authors state about sequence-capture that: "The method generally provides data of higher quality, more consistent loci coverage and subsequently, more accurate SNP calling [17, 30-32]." (p.4, ll.28-30). Regarding their own data, the authors conclude "The methods we tested on *C. finmarchicus* showed that sequence capture enrichment seems to be more adequate for non-model species with large genomes compared to ddRAD-seq" (p.14, ll. 19-21).

I think the authors need to be more cautious, since most of this depends on the experimental design and it is not clear how much this is taken into account. The main problem with the ddRAD approach was that the number of loci was way too large and therefore the coverage per locus and overlap too low. That is a problem of experimental design. The authors state themselves that they used SimRAD to perform an *in silico* digest of the previously sequenced 0.5 part of the genome. This will obviously underestimate the number of expected loci. Did the authors account for that? Please also be more specific if the length of the adapters was accounted for during size selection *in silico* and library prep (it seems like, but please make this explicit for readers). It would be good if the Sim RAD results itself could be provided with the number of expected fragments given the restriction enzymes and size selection and corrected for incompleteness of sequenced genome as supplementary information.

Apart from experimental design, other aspects differ between the approaches that could possibly strongly affect variant detection (which seems to be the main metric the authors use to gauge the quality of their approaches), such as the read mapper and variant caller. For example, why were bowtie2 and SNVer used for the ddRADseq approach, while bwa and GATK were used for the targeted capture (pooled vs. individual sequencing)? If the authors aim to compare the suitability of the two methods, apart from the main conceptual differences, the data should be treated the same way. In other words, please control as much as possible for confounding factors when comparing the two methods.

Another major confounding factor is sequencing effort. I am glad that the authors acknowledge this issue "...however, unequal sequencing effort for the two methods could bias this estimation." (p. 12, ll. 47-49). I think "could" is way too weak here. It almost definitely has an impact, as the authors also acknowledge. The ratio of detected SNPs from capture-based to ddRADseq goes actually down to 7:1 when correcting for sequencing effort. This is still a lot, but much less than the staggering 180:1 that seem to form the basis of what is reported in the abstract. Only the corrected numbers make sense. Please adjust the numbers throughout the manuscript.

Finally, not even the same individuals/populations were used for both approaches (Table 1). That itself makes any sort of quantitative comparisons between the two methods questionable. Generally, I simply think that a well-designed RADseq study is just as valuable and will provide better data than a badly designed sequence-capture study.

Minor comments:

Title: Consider changing to "Towards population genomics in non-model species with large genomes; a case study of the marine zooplankton *Calanus finmarchicus*"

p.3, ll.43: "The double digest RAD-seq, a modification of RAD-seq better suited for species with large genomes, ...". This makes it sound as if ddRAD-seq was invented specifically for large genomes, while it really is generally more flexible and robust independent of genome size.

Review form: Reviewer 2

Is the manuscript scientifically sound in its present form?

Yes

Are the interpretations and conclusions justified by the results?

Yes

Is the language acceptable?

No

Is it clear how to access all supporting data?

No

Do you have any ethical concerns with this paper?

No

Have you any concerns about statistical analyses in this paper?

No

Recommendation?

Major revision is needed (please make suggestions in comments)

Comments to the Author(s)

Review of Choquet et al. "Towards population genomics in non-model species with large genome, case of the marine zooplankton *Calanus finmarchicus*"

Overall.

Choquet et al. represents an interesting examination of two reduced representation methods for *Calanus* species in the Atlantic. Both the species and topic are extremely important and certainly worthy of publication. However, I think the appeal of this work to a non-technical audience could be improved with some additional analyses and perhaps the movement of some of the methods to the supplement. Overall, I think this paper could be suitable for publication following revision, and re-review.

General Comments

1. Use of language is problematic particularly in the introduction. The work could benefit from a solid editing. Some of the issues have been noted but many have not.
2. Overall the paper is largely technical in nature, and perhaps of limited appeal to a non-technical audience. It would have helped if some population genetic results and inferences could be included using the populations analyzed. This would certainly go a long way to increasing the novelty of the contribution and general appeal. The scale of sampling seems sufficient to do something of this type.
3. The methods take a substantial amount of room. Again as this is not a technical journal, I would suggest that much of this is moved to the supplement and referenced within the main document.

Abstract

Ln 14. "Key" is vague, perhaps be more concise here.

Ln 15. Genome reduction is also unclear. Maybe use "reduced representation"

Ln 26. The conclusion that capture is more efficient is ofcourse partially dependent on enzyme choice. Might want to be careful how this is presented.

Introduction

Page 3 Ln 10. "May not" reflect

Page 3 Ln 24. "...consist of"

Page 3 Ln 35. Sentence is awkwardly worded. Please rephrase

Page 4. Ln 14. Consists of

Page 4. Ln 15. Delete "the approach of"

Page 4. Ln 21. Rephrase. "besides, the method.."

Page 4. Ln 10. Why not compare the two approaches? Information on relative cost, time would also be useful. Why was individual RAD not attempted? Was DNA quantity or quality limiting?

Results

Page 11. Ln 2. Some of the population results should be included here such FSTs etc, same below for sequence capture. Also when possible species comparison could be made with overlapping loci.

Discussion

Have other studies attempted population genomics in other marine copepods? If so they should be cited and discussed here.

Review form: Reviewer 3 (Alexander Nater)

Is the manuscript scientifically sound in its present form?

No

Are the interpretations and conclusions justified by the results?

No

Is the language acceptable?

Yes

Is it clear how to access all supporting data?

No

Do you have any ethical concerns with this paper?

No

Have you any concerns about statistical analyses in this paper?

No

Recommendation?

Major revision is needed (please make suggestions in comments)

Comments to the Author(s)

This paper by Choquet and colleagues describes two different approaches to generate genomic-scale SNP data in a non-model organism with a large genome size. The authors employed a pooled ddRAD-seq and a sequence capture enrichment strategy and compared the number of identified SNPs for the two methods. They obtained only very few usable SNPs after coverage filtering in their ddRAD approach and conclude that sequence capture-based methods are a superior strategy to obtain genomic resources in non-model organisms with large genome sizes and without the availability of high quality reference genomes. I have multiple issues with the way the experiments have been set up and the methods are compared:

This is an entirely methodological paper that is missing any kind of biological results. The only results presented are the raw number of SNPs obtained with each approach, without any further validation or application of these marker sets. I would have expected at least some practical application of the two SNP sets to illustrate their usefulness to tackle the questions introduced at the beginning, i.e. analysis of population structure and gene flow (STRUCTURE, PCA, SFS-based demographic modeling, etc.).

The chosen approach of combining ddRAD-seq with Pool-seq seems very strange for the stated goal of detecting population structure, as it necessitates identifying population units before pooling. This makes it impossible to delineate hidden population structure or use individual-based approaches for analyzing population structure and gene flow. It is not clear from the description in the paper if this choice of sequencing method was driven primarily by insufficient DNA quantity obtained from each individual or by economic considerations. Moreover, Pool-seq critically relies on a large number of individuals in a given pool for accurate allele frequency estimation. The authors only used 16 individuals per pool and aimed for a total coverage of 200x, which represents a very modest sequencing effort. Due to the lack of a good reference genome for *in silico* digestion, they ended up with a much higher number of loci with completely insufficient coverage for accurate SNP calling. Not surprisingly, only a very small number of SNPs could be called after applying a coverage filter. To solve this issue, the authors could have sequenced their libraries to higher depth, which would have easily resulted in a much higher number of useable SNPs. Therefore, the results from the ddRAD-seq approach are far from conclusive regarding the usefulness of the approach in this specific setting.

Even though the authors state that they are not aiming at directly comparing the two approaches, they still build most of their discussion around such a comparison of methods. However, their conclusions regarding the superiority of a sequence capture-based approach are not convincing due to the issues with their ddRAD approach mentioned above. Given these shortcomings, I would recommend to completely remove the ddRAD-seq part and instead focus on providing an in-depth description of a sequence capture approach to obtain genomic data in non-model organisms with large genome sizes. For such a methodological paper, however, it is critical that detailed protocols are provided as supporting information and custom analysis scripts are made publicly available. Moreover, the authors should address the problem of non-random selection of SNPs by sequence capture in the discussion. Most SNPs are expected to be within or close to genes when capturing with transcriptome-based baits, which means that selection might influence downstream population genetic analyses. Furthermore, the authors should provide a proof of concept by applying their SNP set to some commonly used analysis methods.

Decision letter (RSOS-180608.R0)

29-May-2018

Dear Mr Choquet,

The editors assigned to your paper ("Towards population genomics in non-model species with large genome, case of the marine zooplankton *Calanus finmarchicus*") have now received comments from reviewers.

Overall, the reviewers are positive about the work, but raise a number of substantive issues surrounding methodological and technical aspects of the study. They wish to see additional data that would reflect the biological application of the methods proposed. In summary, they all agree that major revision of the paper is required.

We would like you to revise your paper in accordance with the referee and Associate Editor suggestions which can be found below (not including confidential reports to the Editor). Please note this decision does not guarantee eventual acceptance.

Please submit a copy of your revised paper within three weeks (i.e. by the 21-Jun-2018). If we do not hear from you within this time then it will be assumed that the paper has been withdrawn. In exceptional circumstances, extensions may be possible if agreed with the Editorial Office in advance. We do not allow multiple rounds of revision so we urge you to make every effort to fully address all of the comments at this stage. If deemed necessary by the Editors, your manuscript will be sent back to one or more of the original reviewers for assessment. If the original reviewers are not available, we may invite new reviewers.

- Data accessibility

It is a condition of publication that all supporting data are made available either as supplementary information or preferably in a suitable permanent repository. The data accessibility section should state where the article's supporting data can be accessed. This section should also include details, where possible of where to access other relevant research materials

such as statistical tools, protocols, software etc can be accessed. If the data have been deposited in an external repository this section should list the database, accession number and link to the DOI for all data from the article that have been made publicly available. Data sets that have been deposited in an external repository and have a DOI should also be appropriately cited in the manuscript and included in the reference list.

If you wish to submit your supporting data or code to Dryad (<http://datadryad.org/>), or modify your current submission to dryad, please use the following link:
<http://datadryad.org/submit?journalID=RSOS&manu=RSOS-180608>

- **Competing interests**

- **Authors' contributions**

- **Acknowledgements**

- **Funding statement**

Please note that Royal Society Open Science will introduce article processing charges for all new submissions received from 1 January 2018. Charges will also apply to papers transferred to Royal Society Open Science from other Royal Society Publishing journals, as well as papers submitted as part of our collaboration with the Royal Society of Chemistry (<http://rsos.royalsocietypublishing.org/chemistry>). If your manuscript is submitted and accepted for publication after 1 Jan 2018, you will be asked to pay the article processing charge, unless you request a waiver and this is approved by Royal Society Publishing. You can find out more about the charges at <http://rsos.royalsocietypublishing.org/page/charges>. Should you have any queries, please contact openscience@royalsociety.org.

Kind regards,
Andrew Dunn
Royal Society Open Science
openscience@royalsociety.org

on behalf of Dr Kristina Sefc (Associate Editor) and Steve Brown (Subject Editor)
openscience@royalsociety.org

Associate Editor's comments (Dr Kristina Sefc):

Associate Editor: 1

Comments to the Author:

The manuscript has been by three reviewers. Overall, they are positive about the study, but agree in their major issues. Reviewers 1 and 2 have serious concerns about the experimental design regarding the comparisons between the SNP identification methods (to the extent that reviewer 3 advises to drop the ddRAD-seq approach altogether from the manuscript), and suggest to focus on biological rather than technical aspects. Similarly, Reviewer 2 suggests to tone down the technical aspect and include biological information. I ask the authors to consider the reviewers' points very thoroughly and revise their manuscript accordingly.

Comments to Author:

Reviewers' Comments to Author:

Reviewer: 1

Comments to the Author(s)

Generating genomic resources for the research community to conduct more powerful population genetic studies in non-model organisms is certainly an important task. In this study, the authors present their workflow to generate a panel of genome-wide markers in the marine copepod *Calanus finmarchicus*, which has been hampered so far by the relatively large genome size of this organism. The authors compare the suitability of two commonly used reduced-representation techniques, ddRAD and targeted-capture. Generally, I think this study has merit, but there are currently too many confounding factors for a meaningful comparison. Most of the differences between number of SNPs found with both methods stem probably from a bad/difficult design, together with differences in sequencing effort, mapping and variant calling, and choice of individuals/populations. I would advise the authors to tone down the comparative aspect, and instead address some of the biological questions that have motivated this study in the first place. After all, the authors set out to gauge the suitability of both methods in order to be able to address questions about the population structure/connectivity of this ecologically important organism. Showing some of those results and comparing them to previous studies might help to demonstrate how important a genome-wide perspective is and give this manuscript more impact. In my opinion, given the many caveats, the main message and conclusions about the suitability of one method over the other is not strongly supported and apart from a fairer treatment throughout the manuscript, I would like to see results on the biological questions before I can recommend this manuscript for publication. That being said, I am happy that the decision whether the slight methodological modifications to the "standard" procedures are sufficient to publish this manuscript in its present form as a method/protocol lies with the editor.

Major comments:

1.) Main message and addressing the biological questions

I think it would help if the authors could make it more clear what the main rationale and message of the manuscript is. For example, in the abstract the authors clearly and directly compare the two methods, but in the last paragraph of the introduction (p. 5, ll. 9-11), the authors state “The present paper does not aim to directly compare the two approaches, ...”. Moreover, the authors highlight the biological questions (e.g. population connectivity, structure) that their approaches are supposed to help with in the introduction (p.4, ll. 49-51), but no analyses / results for this are shown. Based on the sampling, I would think that the authors could have easily addressed some of the biological aspects (e.g. show a PCA, Admixture plot of all populations and compare it to previous studies), which would help to gauge if the genome-wide approach is really more suited and needed to study population structure in marine arthropods. I note, that there is hardly any doubt that a genome-wide approach is superior, it would just be nice if the authors could show this. At the moment it almost appears as if the authors want to simply double their scientific output by having a purely methodological paper now, while saving the results for another manuscript. I would strongly advise against that, unless it can be justified why this is done (e.g. more data is collected at the moment from more populations, whereas this is just a pilot study to see if the method works, etc.).

2.) Comparison of the two approaches and confounding factors

The authors make it sound as if ddRADseq is generally worse than the capture-based approach. For example, already in the introduction the authors state about sequence-capture that: “The method generally provides data of higher quality, more consistent loci coverage and subsequently, more accurate SNP calling [17, 30-32].” (p.4, ll.28-30). Regarding their own data, the authors conclude “The methods we tested on *C. finmarchicus* showed that sequence capture enrichment seems to be more adequate for non-model species with large genomes compared to ddRAD-seq” (p.14, ll. 19-21).

I think the authors need to be more cautious, since most of this depends on the experimental design and it is not clear how much this is taken into account. The main problem with the ddRAD approach was that the number of loci was way too large and therefore the coverage per locus and overlap too low. That is a problem of experimental design. The authors state themselves that they used SimRAD to perform an in silico digest of the previously sequenced 0.5 part of the genome. This will obviously underestimate the number of expected loci. Did the authors account for that? Please also be more specific if the length of the adapters was accounted for during size selection in silico and library prep (it seems like, but please make this explicit for readers). It would be good if the Sim RAD results itself could be provided with the number of expected fragments given the restriction enzymes and size selection and corrected for incompleteness of sequenced genome as supplementary information.

Apart from experimental design, other aspects differ between the approaches that could possibly strongly affect variant detection (which seems to be the main metric the authors use to gauge the quality of their approaches), such as the read mapper and variant caller. For example, why were bowtie2 and SNVer used for the ddRADseq approach, while bwa and GATK were used for the targeted capture (pooled vs. individual sequencing)? If the authors aim to compare the suitability of the two methods, apart from the main conceptual differences, the data should be treated the same way. In other words, please control as much as possible for confounding factors when comparing the two methods.

Another major confounding factor is sequencing effort. I am glad that the authors acknowledge this issue “...however, unequal sequencing effort for the two methods could bias this estimation.” (p. 12, ll. 47-49). I think “could” is way too weak here. It almost definitely has an impact, as the authors also acknowledge. The ratio of detected SNPs from capture-based to ddRADseq goes actually down to 7:1 when correcting for sequencing effort. This is still a lot, but

much less than the staggering 180:1 that seem to form the basis of what is reported in the abstract. Only the corrected numbers make sense. Please adjust the numbers throughout the manuscript.

Finally, not even the same individuals/populations were used for both approaches (Table 1). That itself makes any sort of quantitative comparisons between the two methods questionable. Generally, I simply think that a well-designed RADseq study is just as valuable and will provide better data than a badly designed sequence-capture study.

Minor comments:

Title: Consider changing to "Towards population genomics in non-model species with large genomes; a case study of the marine zooplankton *Calanus finmarchicus*"

p.3, ll.43: "The double digest RAD-seq, a modification of RAD-seq better suited for species with large genomes, ...". This makes it sound as if ddRAD-seq was invented specifically for large genomes, while it really is generally more flexible and robust independent of genome size.

Reviewer: 2

Comments to the Author(s)

Review of Choquet et al. "Towards population genomics in non-model species with large genome, case of the marine zooplankton *Calanus finmarchicus*"

Overall.

Choquet et al. represents an interesting examination of two reduced representation methods for *Calanus* species in the Atlantic. Both the species and topic are extremely important and certainly worthy of publication. However I think the appeal of this work to a non-technical audience could be improved with some additional analyses and perhaps the movement of some of the methods to the supplement. Overall, I think this paper could be suitable for publication following revision, and re-review.

General Comments

1. Use of language is problematic particularly in the introduction. The work could benefit from a solid editing. Some of the issues have been noted but many have not.

2. Overall the paper is largely technical in nature, and perhaps of limited appeal to a non-technical audience. It would have helped if some population genetic results and inferences could be included using the populations analyzed. This would certainly go a long way to increasing the novelty of the contribution and general appeal. The scale of sampling seems sufficient to do something of this type.

3. The methods take a substantial amount of room. Again as this is not a technical journal, I would suggest that much of this is moved to the supplement and referenced within the main document.

Abstract

Ln 14. "Key" is vague, perhaps be more concise here.

Ln 15. Genome reduction is also unclear. Maybe use "reduced representation"

Ln 26. The conclusion that capture is more efficient is of course partially dependent on enzyme choice. Might want to be careful how this is presented.

Introduction

Page 3 Ln 10. "May not" reflect

Page 3 Ln 24. "...consist of"

Page 3 Ln 35. Sentence is awkwardly worded. Please rephrase

Page 4. Ln 14. Consists of

Page 4. Ln 15. Delete "the approach of"

Page 4. Ln 21. Rephrase. "besides, the method.."

Page 4. Ln 10. Why not compare the two approaches? Information on relative cost, time would also be useful. Why was individual RAD not attempted? Was DNA quantity or quality limiting?

Results

Page 11. Ln 2. Some of the population results should be included here such FSTs etc, same below for sequence capture. Also when possible species comparison could be made with overlapping loci.

Discussion

Have other studies attempted population genomics in other marine copepods? If so they should be cited and discussed here.

Reviewer: 3

Comments to the Author(s)

This paper by Choquet and colleagues describes two different approaches to generate genomic-scale SNP data in a non-model organism with a large genome size. The authors employed a pooled ddRAD-seq and a sequence capture enrichment strategy and compared the number of identified SNPs for the two methods. They obtained only very few usable SNPs after coverage filtering in their ddRAD approach and conclude that sequence capture-based methods are a superior strategy to obtain genomic resources in non-model organisms with large genome sizes and without the availability of high quality reference genomes. I have multiple issues with the way the experiments have been set up and the methods are compared:

This is an entirely methodological paper that is missing any kind of biological results. The only results presented are the raw number of SNPs obtained with each approach, without any further validation or application of these marker sets. I would have expected at least some practical application of the two SNP sets to illustrate their usefulness to tackle the questions introduced at the beginning, i.e. analysis of population structure and gene flow (STRUCTURE, PCA, SFS-based demographic modeling, etc.).

The chosen approach of combining ddRAD-seq with Pool-seq seems very strange for the stated goal of detecting population structure, as it necessitates identifying population units before pooling. This makes it impossible to delineate hidden population structure or use individual-based approaches for analyzing population structure and gene flow. It is not clear from the description in the paper if this choice of sequencing method was driven primarily by insufficient DNA quantity obtained from each individual or by economic considerations. Moreover, Pool-seq critically relies on a large number of individuals in a given pool for accurate allele frequency estimation. The authors only used 16 individuals per pool and aimed for a total coverage of 200x, which represents a very modest sequencing effort. Due to the lack of a good reference genome for *in silico* digestion, they ended up with a much higher number of loci with completely insufficient

coverage for accurate SNP calling. Not surprisingly, only a very small number of SNPs could be called after applying a coverage filter. To solve this issue, the authors could have sequenced their libraries to higher depth, which would have easily resulted in a much higher number of useable SNPs. Therefore, the results from the ddRAD-seq approach are far from conclusive regarding the usefulness of the approach in this specific setting.

Even though the authors state that they are not aiming at directly comparing the two approaches, they still build most of their discussion around such a comparison of methods. However, their conclusions regarding the superiority of a sequence capture-based approach are not convincing due to the issues with their ddRAD approach mentioned above. Given these shortcomings, I would recommend to completely remove the ddRAD-seq part and instead focus on providing an in-depth description of a sequence capture approach to obtain genomic data in non-model organisms with large genome sizes. For such a methodological paper, however, it is critical that detailed protocols are provided as supporting information and custom analysis scripts are made publicly available. Moreover, the authors should address the problem of non-random selection of SNPs by sequence capture in the discussion. Most SNPs are expected to be within or close to genes when capturing with transcriptome-based baits, which means that selection might influence downstream population genetic analyses. Furthermore, the authors should provide a proof of concept by applying their SNP set to some commonly used analysis methods.

Author's Response to Decision Letter for (RSOS-180608.R0)

See Appendix A.

RSOS-180608.R1 (Revision)

Review form: Reviewer 1

Is the manuscript scientifically sound in its present form?

Yes

Are the interpretations and conclusions justified by the results?

Yes

Is the language acceptable?

Yes

Is it clear how to access all supporting data?

Yes

Do you have any ethical concerns with this paper?

No

Have you any concerns about statistical analyses in this paper?

No

Recommendation?

Accept with minor revision (please list in comments)

Comments to the Author(s)

The authors have addressed the main issues of the manuscript by moving the ddRAD-seq part to the supplementary information and providing biological analyses. I am happy to say that I think this manuscript is therefore now suitable for publication. There are a few things that the authors might want to edit/correct, but I'm sure they can be revised easily. I congratulate the authors on their nice piece of work and look forward to seeing more pop genomics studies on marine arthropods published.

Minor comments:

1.) ll.117-120: "The enzymes for the digestion were chosen after in silico digestion of the very small fraction of the genome sequenced so far (< 0.5 %), resulting in a very high number of fragments, requiring a costly sequencing effort in order to achieve sufficient coverage."

I think the authors want to make the point that the number of obtained fragments was much higher than anticipated from their in silico digest here, but I don't think this really becomes clear and I don't understand why this would be the case (when properly done and extrapolated from the 0.5%). The authors themselves state in their response letter that according to the simRAD manual this is apparently not expected. I think the body size/ minimum DNA requirement is a much stronger argument and I would advise the authors to focus on that and say that they decided to go with targeted sequencing mainly because of that or provide a conclusive explanation why exactly their ddRAD-seq design failed despite the in silico digest.

2.) ll. 372-376: Comparing heterozygosity levels at biallelic SNPs and (likely) multiallelic msats is flawed. The highest possible het. level based on the former is 0.5, whereas msats with more than two alleles will almost always show higher het. levels. Please revise.

3.) Does Figure 2 really make an important point that has to be in the main article? This seems to me more like supplementary information.

4.) Can you please make the color code for the different pops in Fig. 4 more distinct? It's hard to distinguish the pops at the moment.

Typos:

- l. 67: appear
- l. 85: targeted sequences
- l. 92: Capture probes sets have also proofed
- l. 93: delete "on" in "reported on"
- l. 125: "obtain an informative"
- l. 331: "is critical, but"
- l. 384: "shows"
- l. 417: "at the Canadian east coast"
- l. 419: "...detected any hybrids in any of the 85 locations..., though."

Review form: Reviewer 3 (Alexander Nater)

Is the manuscript scientifically sound in its present form?

No

Are the interpretations and conclusions justified by the results?

Yes

Is the language acceptable?

Yes

Is it clear how to access all supporting data?

Yes

Do you have any ethical concerns with this paper?

No

Have you any concerns about statistical analyses in this paper?

Yes

Recommendation?

Major revision is needed (please make suggestions in comments)

Comments to the Author(s)

Alexander Nater

Department of Biology, University of Konstanz, Germany

The authors have revised their manuscript substantially and implemented my own and the other reviewers' recommendations regarding the removal of the ddRAD-seq part from the main text and the incorporation of some biological applications of their new SNP sets. As a result, the paper is now more appealing for a broad, non-technical readership, even though the population genetic analyses remain rather superficial. I have some methodological concerns concerning the population genetics part of the manuscript that should be carefully addressed before I can recommend the paper for publication.

Line 68-69: It's not clear what the relationship between RAD-seq and allele-specific expression is. I guess the authors refer to problems with RNA-seq based SNP detection methods, but this should be clarified.

Line 93: "Capture enrichments approaches have ..."

Line 114-126: This paragraph is a bit awkward to read and needs some polishing.

Line 231: "and mapped reads were realigned ..."

Lines 234-236: Not clear if the SNP calling was performed for all individuals for each species or for both species together.

Line 248-252: It doesn't make sense to apply a HWE filter to the SNPs if there is potential population structure in the data set, especially if the SNPs are later supposed to be used to test for such population structure. Moreover, applying minor allele frequency filters to the data will strongly bias the estimates of individual heterozygosity and nucleotide diversity. Such variant-specific filters should be avoided if the goal is to obtain accurate estimates of genetic diversity. In such cases, suitable filters should be applicable to both variant and non-variant sites, such as individual read depth, missingness, etc.

Line 249-250: Expected heterozygosity is a population statistic and not applicable at the individual level.

Line 251-252: It doesn't make sense to calculate nucleotide diversity on such small windows if the main goal is to compare genome-wide diversity among populations and species. Such small windows will only contain a very low number of variant sites and the window-wise estimates will be extremely noisy. A better approach would be to calculate nucleotide diversity across the entire assembly length and then use a resampling procedure to obtain confidence intervals. Moreover, it's not clear from the description how the authors obtained the proper number of callable yet non-variant sites within each window. For accurate estimates of nucleotide diversity, it's important to distinguish between well covered (i.e. callable) non-variant sites and sites that are not sufficiently covered (i.e. non-callable sites with unknown variant status). This can be achieved quite easily by emitting both confident variant and non-variant sites during variant calling with GATK.

Lines 298-300: I guess these values are the mean individual heterozygosities at variant sites, not the proportion of heterozygous sites?

Lines 309-311: Provide p-values for the global F_{st} estimates.

Lines 311-313: How do you explain the outliers for *C. Finmarchicus* in Fig. 4B? Are there any apparent quality differences compared to the other individuals that could point to technical reasons for this?

Line 313: "..., consistent with ..."

Lines 331-332: It's not clear why the occurrence of high gene flow would necessitate a genomic approach.

Lines 369-370: Tone down this statement. The selected approach for detecting outliers with BayeScan using only exonic SNPs is not suitable to reliably assess neutrality. At best, you can detect loci under recent and strong positive selection. I would like to see a site frequency spectrum of all SNPs for the two species in the Supplementary Materials to get a better impression of potential selective constraints affecting these SNPs.

Lines 371-373: It's not clear how these heterozygosities have been calculated. Were they averaged over all positions in the two data sets that were polymorphic relative to the reference assembly? If so, there is an obvious downwards bias for *C. glacialis*, as this species would probably have quite a few fixed differences to the *C. finmarchicus* reference. These sites would have a mean individual heterozygosity of 0, as they are not polymorphic in *C. glacialis*. Anyways, SNP heterozygosity is not a suitable statistic to compare levels of genetic diversity between populations or species and should not be compared to heterozygosities for highly variable markers such as microsatellites. The proper statistic to assess genetic diversity is nucleotide diversity.

Lines 394-395: A larger number of markers will lead to higher precision, not necessarily higher accuracy.

Line 403-404: Again, SNP heterozygosity is not a proper statistic to measure genetic diversity. If you want to compare genetic diversity across studies and species, nucleotide diversity should be used. For example, a population expansion in *Centropages typicus* could easily explain the lower levels of SNP heterozygosity despite similar or higher levels of genetic diversity.

Table 2: Also provide mean sequencing depth for each individual after filtering.

Fig. 3: "Individual heterozygosity" instead of "Expected heterozygosity".

Decision letter (RSOS-180608.R1)

24-Jul-2018

Dear Mr Choquet:

Manuscript ID RSOS-180608.R1 entitled "Towards population genomics in non-model species with large genomes; a case study of the marine zooplankton *Calanus finmarchicus*" which you submitted to Royal Society Open Science, has been reviewed. The comments of the reviewer(s) are included at the bottom of this letter.

One reviewer is very positive about publication now, commenting that the paper is ready for publication. The other reviewer however recommends some additional methodological concerns surround the population genetics which you will need to address before the manuscript is acceptable for publication.

Please submit a copy of your revised paper before 16-Aug-2018. Please note that the revision deadline will expire at 00.00am on this date. If we do not hear from you within this time then it will be assumed that the paper has been withdrawn. In exceptional circumstances, extensions may be possible if agreed with the Editorial Office in advance. We do not allow multiple rounds of revision so we urge you to make every effort to fully address all of the comments at this stage. If deemed necessary by the Editors, your manuscript will be sent back to one or more of the original reviewers for assessment. If the original reviewers are not available we may invite new reviewers.

- Ethics statement

- Data accessibility

- Competing interests

- Authors' contributions

- Acknowledgements

- Funding statement

Please note that Royal Society Open Science charge article processing charges for all new submissions that are accepted for publication. Charges will also apply to papers transferred to Royal Society Open Science from other Royal Society Publishing journals, as well as papers submitted as part of our collaboration with the Royal Society of Chemistry (<http://rsos.royalsocietypublishing.org/chemistry>). If your manuscript is newly submitted and subsequently accepted for publication, you will be asked to pay the article processing charge, unless you request a waiver and this is approved by Royal Society Publishing. You can find out more about the charges at <http://rsos.royalsocietypublishing.org/page/charges>. Should you have any queries, please contact openscience@royalsociety.org.

on behalf of Dr Kristina Sefc (Associate Editor) and Prof. Steve Brown (Subject Editor)
 openscience@royalsociety.org

Reviewer comments to Author:
 Reviewer: 1

Comments to the Author(s)

The authors have addressed the main issues of the manuscript by moving the ddRAD-seq part to the supplementary information and providing biological analyses. I am happy too say that I think this manuscript is therefore now suitable for publication. There are a few things that the authors might want to edit/correct, but I'm sure they can be revised easily. I congratulate the authors on their nice piece of work and look forward to seeing more pop genomics studies on marine arthropods published.

Minor comments:

- 1.) ll.117-120: "The enzymes for the digestion were chosen after in silico digestion of the very small fraction of the genome sequenced so far (< 0.5 %), resulting in a very high number of fragments, requiring a costly sequencing effort in order to achieve sufficient coverage."
 I think the authors want to make the point that the number of obtained fragments was much higher than anticipated from their in silico digest here, but I don't think this really becomes clear and I don't understand why this would be the case (when properly done and extrapolated from the 0.5%). The authors themselves state in their response letter that according to the simRAD manual this is apparently not expected. I think the body size/ minimum DNA requirement is a much stronger argument and I would advise the authors to focus on that and say that they decided to go with targeted sequencing mainly because of that or provide a conclusive explanation why exactly their ddRAD-seq design failed despite the in silico digest.
- 2.) ll. 372-376: Comparing heterozygosity levels at biallelic SNPs and (likely) multiallelic msats is flawed. The highest possible het. level based on the former is 0.5, whereas msats with more than two alleles will almost always show higher het. levels. Please revise.
- 3.) Does Figure 2 really make an important point that has to be in the main article? This seems to me more like supplementary information.
- 4.) Can you please make the color code for the different pops in Fig. 4 more distinct? It's hard to distinguish the pops at the moment.

Typos:

- l. 67: appear
- l. 85: targeted sequences

- l. 92: Capture probes sets have also proofed
- l. 93: delete "on" in "reported on"
- l. 125: "obtain an informative"
- l. 331: "is critical, but"
- l. 384: "shows"
- l. 417: "at the Canadian east coast"
- l. 419: "...detected any hybrids in any of the 85 locations..., though."

Reviewer: 3

Comments to the Author(s)

Alexander Nater

Department of Biology, University of Konstanz, Germany

The authors have revised their manuscript substantially and implemented my own and the other reviewers' recommendations regarding the removal of the ddRAD-seq part from the main text and the incorporation of some biological applications of their new SNP sets. As a result, the paper is now more appealing for a broad, non-technical readership, even though the population genetic analyses remain rather superficial. I have some methodological concerns concerning the population genetics part of the manuscript that should be carefully addressed before I can recommend the paper for publication.

Line 68-69: It's not clear what the relationship between RAD-seq and allele-specific expression is. I guess the authors refer to problems with RNA-seq based SNP detection methods, but this should be clarified.

Line 93: "Capture enrichments approaches have ..."

Line 114-126: This paragraph is a bit awkward to read and needs some polishing.

Line 231: "and mapped reads were realigned ..."

Lines 234-236: Not clear if the SNP calling was performed for all individuals for each species or for both species together.

Line 248-252: It doesn't make sense to apply a HWE filter to the SNPs if there is potential population structure in the data set, especially if the SNPs are later supposed to be used to test for such population structure. Moreover, applying minor allele frequency filters to the data will strongly bias the estimates of individual heterozygosity and nucleotide diversity. Such variant-specific filters should be avoided if the goal is to obtain accurate estimates of genetic diversity. In such cases, suitable filters should be applicable to both variant and non-variant sites, such as individual read depth, missingness, etc.

Line 249-250: Expected heterozygosity is a population statistic and not applicable at the individual level.

Line 251-252: It doesn't make sense to calculate nucleotide diversity on such small windows if the main goal is to compare genome-wide diversity among populations and species. Such small windows will only contain a very low number of variant sites and the window-wise estimates will be extremely noisy. A better approach would be to calculate nucleotide diversity across the entire assembly length and then use a resampling procedure to obtain confidence intervals. Moreover, it's not clear from the description how the authors obtained the proper number of callable yet non-variant sites within each window. For accurate estimates of nucleotide diversity,

it's important to distinguish between well covered (i.e. callable) non-variant sites and sites that are not sufficiently covered (i.e. non-callable sites with unknown variant status). This can be achieved quite easily by emitting both confident variant and non-variant sites during variant calling with GATK.

Lines 298-300: I guess these values are the mean individual heterozygosities at variant sites, not the proportion of heterozygous sites?

Lines 309-311: Provide p-values for the global F_{st} estimates.

Lines 311-313: How do you explain the outliers for *C. finmarchicus* in Fig. 4B? Are there any apparent quality differences compared to the other individuals that could point to technical reasons for this?

Line 313: "..., consistent with ..."

Lines 331-332: It's not clear why the occurrence of high gene flow would necessitate a genomic approach.

Lines 369-370: Tone down this statement. The selected approach for detecting outliers with BayeScan using only exonic SNPs is not suitable to reliably assess neutrality. At best, you can detect loci under recent and strong positive selection. I would like to see a site frequency spectrum of all SNPs for the two species in the Supplementary Materials to get a better impression of potential selective constraints affecting these SNPs.

Lines 371-373: It's not clear how these heterozygosities have been calculated. Were they averaged over all positions in the two data sets that were polymorphic relative to the reference assembly? If so, there is an obvious downwards bias for *C. glacialis*, as this species would probably have quite a few fixed differences to the *C. finmarchicus* reference. These sites would have a mean individual heterozygosity of 0, as they are not polymorphic in *C. glacialis*. Anyways, SNP heterozygosity is not a suitable statistic to compare levels of genetic diversity between populations or species and should not be compared to heterozygosities for highly variable markers such as microsatellites. The proper statistic to assess genetic diversity is nucleotide diversity.

Lines 394-395: A larger number of markers will lead to higher precision, not necessarily higher accuracy.

Line 403-404: Again, SNP heterozygosity is not a proper statistic to measure genetic diversity. If you want to compare genetic diversity across studies and species, nucleotide diversity should be used. For example, a population expansion in *Centropages typicus* could easily explain the lower levels of SNP heterozygosity despite similar or higher levels of genetic diversity.

Table 2: Also provide mean sequencing depth for each individual after filtering.

Fig. 3: "Individual heterozygosity" instead of "Expected heterozygosity".

Author's Response to Decision Letter for (RSOS-180608.R1)

See Appendix B.

RSOS-180608.R2 (Revision)

Review form: Reviewer 3 (Alexander Nater)

Is the manuscript scientifically sound in its present form?

Yes

Are the interpretations and conclusions justified by the results?

Yes

Is the language acceptable?

Yes

Is it clear how to access all supporting data?

No

Do you have any ethical concerns with this paper?

No

Have you any concerns about statistical analyses in this paper?

No

Recommendation?

Accept with minor revision (please list in comments)

Comments to the Author(s)

The paper by Choquet and colleagues went through another round of substantial revisions and the authors have satisfactorily addressed my previous methodological concerns. I only have a few suggestions to improve the description of the applied methods, which still lacks quite a few important details:

Line 255: "... to also output ... in the outputted VCF file.". I would suggest replacing "outputted" by "resulting".

Line 256-257: "we filtered the sites to keep only those covered 5x or more and present in at least 80% of the individuals". The current description sounds as if total coverage per site needed to be $\geq 5x$, rather than coverage per individual. Please clarify what is meant by a site being present in at least 80% of individuals.

Line 260-264: Omit "We considered that the number of individuals was too low to make accurate estimates of nucleotide diversity on a per-site basis, as each site is very limited in the possible values that can be obtained. To get around this,". It is obvious that you do not want to report nucleotide diversity for each site separately. Nonetheless, it is important to clarify that nucleotide diversity was estimated on a per-site basis and averaged in 780-bp windows using only the sites that passed the filtering. The description "we measured diversity in non-overlapping windows of 780 bp length" is confusing here.

Line 264: Mean of π across windows for each population?

Lines 265-268: Same issue as above. The calculation would still be on a per-SNP level, but you only report genome-wide averages per individual.

Line 266-267: Reporting average observed heterozygosities per individual is fine. My point was referring to the term “individual expected heterozygosity”. Expected heterozygosity is a population-level statistic, it simply cannot be calculated on a per-individual basis. What you are calculating here is observed heterozygosity, i.e. you are calculating the proportion of heterozygous SNP for each individual. Expected heterozygosity would be the expected proportion of heterozygous genotypes given the estimated allele frequencies in a population.

Line 276: “variants phased” instead of “haplotypes phased”

Line 278: “Variants present in less than 80% of genotypes were filtered out.” It is not clear what this is supposed to mean. It reads as if SNPs with less than 80% of genotypes carrying the alternative allele were filtered out, but that would not make much sense at all.

Line 292: Custom scripts should either be provided in the Supplementary Materials or uploaded to a suitable repository.

Lines 427-434: Comparing absolute values of nucleotide diversity between autosomal SNPs and mitochondrial genes is quite pointless, as nucleotide diversity depends on the mutation rate, which is obviously different between these marker types.

Line 451: “(PC1: ~12% of variance explained).”

Decision letter (RSOS-180608.R2)

30-Nov-2018

Dear Mr Choquet:

On behalf of the Editors, I am pleased to inform you that your Manuscript RSOS-180608.R2 entitled "Towards population genomics in non-model species with large genomes; a case study of the marine zooplankton *Calanus finmarchicus*" has been accepted for publication in Royal Society Open Science subject to minor revision in accordance with the referee suggestions. Please find the referees' comments at the end of this email.

The reviewers and Subject Editor have recommended publication, but also suggest some minor revisions to your manuscript. Therefore, I invite you to respond to the comments and revise your manuscript.

- Ethics statement

- Data accessibility

It is a condition of publication that all supporting data are made available either as supplementary information or preferably in a suitable permanent repository. The data accessibility section should state where the article's supporting data can be accessed. This section should also include details, where possible of where to access other relevant research materials

such as statistical tools, protocols, software etc can be accessed. If the data has been deposited in an external repository this section should list the database, accession number and link to the DOI for all data from the article that has been made publicly available. Data sets that have been deposited in an external repository and have a DOI should also be appropriately cited in the manuscript and included in the reference list.

If you wish to submit your supporting data or code to Dryad (<http://datadryad.org/>), or modify your current submission to dryad, please use the following link:
<http://datadryad.org/submit?journalID=RSOS&manu=RSOS-180608.R2>

- **Competing interests**

- **Authors' contributions**

- **Acknowledgements**

- **Funding statement**

Because the schedule for publication is very tight, it is a condition of publication that you submit the revised version of your manuscript before 09-Dec-2018. Please note that the revision deadline will expire at 00.00am on this date. If you do not think you will be able to meet this date please let me know immediately.

Please note that Royal Society Open Science charge article processing charges for all new submissions that are accepted for publication. Charges will also apply to papers transferred to Royal Society Open Science from other Royal Society Publishing journals, as well as papers submitted as part of our collaboration with the Royal Society of Chemistry (<http://rsos.royalsocietypublishing.org/chemistry>). If your manuscript is newly submitted and subsequently accepted for publication, you will be asked to pay the article processing charge, unless you request a waiver and this is approved by Royal Society Publishing. You can find out more about the charges at <http://rsos.royalsocietypublishing.org/page/charges>. Should you have any queries, please contact openscience@royalsociety.org.

on behalf of Dr Kristina Sefc (Associate Editor) and Professor Steve Brown (Subject Editor)
openscience@royalsociety.org

Reviewer comments to Author:

Reviewer: 3

Comments to the Author(s)

The paper by Choquet and colleagues went through another round of substantial revisions and the authors have satisfactorily addressed my previous methodological concerns. I only have a few suggestions to improve the description of the applied methods, which still lacks quite a few important details:

Line 255: "... to also output ... in the outputted VCF file.". I would suggest replacing "outputted" by "resulting".

Line 256-257: "we filtered the sites to keep only those covered 5x or more and present in at least 80% of the individuals". The current description sounds as if total coverage per site needed to be $\geq 5x$, rather than coverage per individual. Please clarify what is meant by a site being present in at least 80% of individuals.

Line 260-264: Omit "We considered that the number of individuals was too low to make accurate estimates of nucleotide diversity on a per-site basis, as each site is very limited in the possible values that can be obtained. To get around this,". It is obvious that you do not want to report nucleotide diversity for each site separately. Nonetheless, it is important to clarify that nucleotide diversity was estimated on a per-site basis and averaged in 780-bp windows using only the sites that passed the filtering. The description "we measured diversity in non-overlapping windows of 780 bp length" is confusing here.

Line 264: Mean of π across windows for each population?

Lines 265-268: Same issue as above. The calculation would still be on a per-SNP level, but you only report genome-wide averages per individual.

Line 266-267: Reporting average observed heterozygosities per individual is fine. My point was referring to the term "individual expected heterozygosity". Expected heterozygosity is a population-level statistic, it simply cannot be calculated on a per-individual basis. What you are calculating here is observed heterozygosity, i.e. you are calculating the proportion of heterozygous SNP for each individual. Expected heterozygosity would be the expected proportion of heterozygous genotypes given the estimated allele frequencies in a population.

Line 276: "variants phased" instead of "haplotypes phased"

Line 278: "Variants present in less than 80% of genotypes were filtered out." It is not clear what this is supposed to mean. It reads as if SNPs with less than 80% of genotypes carrying the alternative allele were filtered out, but that would not make much sense at all.

Line 292: Custom scripts should either be provided in the Supplementary Materials or uploaded to a suitable repository.

Lines 427-434: Comparing absolute values of nucleotide diversity between autosomal SNPs and mitochondrial genes is quite pointless, as nucleotide diversity depends on the mutation rate, which is obviously different between these marker types.

Line 451: "(PC1: ~12% of variance explained)."

Author's Response to Decision Letter for (RSOS-180608.R2)

See Appendix C.

Decision letter (RSOS-180608.R3)

07-Jan-2019

Dear Mr Choquet,

I am pleased to inform you that your manuscript entitled "Towards population genomics in non-model species with large genomes; a case study of the marine zooplankton *Calanus finmarchicus*" is now accepted for publication in Royal Society Open Science.

on behalf of Dr Kristina Sefc (Associate Editor) and Steve Brown (Subject Editor)
openscience@royalsociety.org

Appendix A

Dear Editor,

We would like to thank you and the three anonymous reviewers for the constructive comments, which we found very helpful and relevant. We have now revised our manuscript accordingly. We have addressed all the points raised by the reviewers in this new version. Below are our responses to the primary issues raised by you, the associate editor and reviewers, as well as detailed responses to the comments of the reviewers.

Associate Editor's comments (Dr Kristina Sefc):

Reviewers 1 and 2 have serious concerns about the experimental design regarding the comparisons between the SNP identification methods (to the extent that reviewer 3 advises to drop the ddRAD-seq approach altogether from the manuscript), and suggest to focus on biological rather than technical aspects. Similarly, Reviewer 2 suggests to tone down the technical aspect and include biological information. I ask the authors to consider the reviewers' points very thoroughly and revise their manuscript accordingly.

We agree with the opinions from the Associate Editor and the Reviewers and we have removed the ddRAD-seq part from the main text. This is now attached as Supplementary Material. We have changed the focus of the paper accordingly, and provided analyses of the SNPs data.

Comments reviewer 1

I would advise the authors to tone down the comparative aspect, and instead address some of the biological questions that have motivated this study in the first place. After all, the authors set out to gauge the suitability of both methods in order to be able to address questions about the population structure/connectivity of this ecologically important organism. Showing some of those results and comparing them to previous studies might help to demonstrate how important a genome-wide perspective is and give this manuscript more impact. In my opinion, given the many caveats, the main message and conclusions about the suitability of one method over the other is not strongly supported and apart from a fairer treatment throughout the manuscript, I would like to see results on the biological questions before I can

recommend this manuscript for publication. That being said, I am happy that the decision whether the slight methodological modifications to the “standard” procedures are sufficient to publish this manuscript in its present form as a method/protocol lies with the editor.

We have substantially toned down the technical comparative aspect of the manuscript. We have put the ddRAD-seq as Supplementary Material. The focus is now more on how we developed / set up a protocol of genome-reduced representation for Calanus, and we have made several analyses (population structure, genetic diversity, selection) in order to discuss more about the novelty brought by a genome-wide approach to the understanding of Calanus biology.

1.) Main message and addressing the biological questions

I think it would help if the authors could make it more clear what the main rationale and message of the manuscript is. For example, in the abstract the authors clearly and directly compare the two methods, but in the last paragraph of the introduction (p. 5, ll. 9-11), the authors state “The present paper does not aim to directly compare the two approaches, ...”. Moreover, the authors highlight the biological questions (e.g. population connectivity, structure) that their approaches are supposed to help with in the introduction (p.4, ll. 49-51), but no analyses / results for this are shown. Based on the sampling, I would think that the authors could have easily addressed some of the biological aspects (e.g. show a PCA, Admixture plot of all populations and compare it to previous studies), which would help to gauge if the genome-wide approach is really more suited and needed to study population structure in marine arthropods. I note, that there is hardly any doubt that a genome-wide approach is superior, it would just be nice if the authors could show this. At the moment it almost appears as if the authors want to simply double their scientific output by having a purely methodological paper now, while saving the results for another manuscript. I would strongly advise against that, unless it can be justified why this is done (e.g. more data is collected at the moment from more populations, whereas this is just a pilot study to see if the method works, etc.).

We have revised the Abstract. PCAs were done, FST was calculated, tests for selection were done as well as estimation of genetic diversity and nucleotide diversity. The paper is not focused on the methodology only anymore. We discuss the meanings of the new analyses done, and compare them with results from previous studies that used microsatellites.

It is correct that this is just a pilot study to see if the method works, and more data are being collected at the moment from populations spanning the whole distributional ranges of the species in order to use this method on a large dataset.

2.) Comparison of the two approaches and confounding factors

The authors make it sound as if ddRADseq is generally worse than the capture-based approach. For example, already in the introduction the authors state about sequence-capture that: “The method generally provides data of higher quality, more consistent loci coverage and subsequently, more accurate SNP calling [17, 30-32].” (p.4, ll.28-30). Regarding their own data, the authors conclude “The methods we tested on *C. finmarchicus* showed that sequence capture enrichment seems to be more adequate for non-model species with large genomes compared to ddRAD-seq” (p.14, ll. 19-21).

I think the authors need to be more cautious, since most of this depends on the experimental design and it is not clear how much this is taken into account. The main problem with the ddRAD approach was that the number of loci was way too large and therefore the coverage per locus and overlap too low. That is a problem of experimental design. The authors state themselves that they used SimRAD to perform an in silico digest of the previously sequenced 0.5 part of the genome. This will obviously underestimate the number of expected loci. Did the authors account for that? Please also be more specific if the length of the adapters was accounted for during size selection in silico and library prep (it seems like, but please make this explicit for readers). It would be good if the Sim RAD results itself could be provided with the number of expected fragments given the restriction enzymes and size selection and corrected for incompleteness of sequenced genome as supplementary information.

We agree that the design was not optimal for the ddRAD approach, and we justify that this was due to technical challenge of DNA amount requirement of the method, too high to be reached for *Calanus*, hence we had to pool. We now clarified this in the main text, and most of the ddRAD part was moved to Supplementary Material. We accounted for length of adapters during size selection and provided this information in the Supplementary Material 1.

Regarding SIM RAD, we didn't expect that 0.5% of genome will result in underestimation since in the simmered manual is stated: “To limit memory usage and speed up computation time, it is strongly recommended to sub-select a fraction of the available reference DNA contigs. Usually, a proportion of 0.10 % of the reference contigs should be representative enough to give a good idea of the genome characteristics of the species”. But we took into account that analysis is run with 0.5% and multiplied the results accordingly to reach 100% of the genome.

Apart from experimental design, other aspects differ between the approaches that could possibly strongly affect variant detection (which seems to be the main metric the authors use to gauge the quality of their approaches), such as the read mapper and variant caller. For example, why were bowtie2 and SNVer used for the ddRADseq approach, while bwa and GATK were used for the targeted capture (pooled vs. individual sequencing)? If the authors aim to compare the suitability of the two methods, apart from the main conceptual differences, the data should be treated the same way. In other words, please control as much as possible for confounding factors when comparing the two methods.

Data from the two different approaches were not treated the same way, because depending on how they are generated, sequencing data require different programs that have to be appropriate for the corresponding type of data. However, parameters for SNPs filtering were similar. The RAD-seq has now been removed from the main text, and we toned down the comparative aspect.

Another major confounding factor is sequencing effort. I am glad that the authors acknowledge this issue “...however, unequal sequencing effort for the two methods could bias this estimation.” (p. 12, ll. 47-49). I think “could” is way too weak here. It almost definitely has an impact, as the authors also acknowledge. The ratio of detected SNPs from capture-based to ddRADseq goes actually down to 7:1 when correcting for sequencing effort. This is still a lot, but much less than the staggering 180:1 that seem to form the basis of what is reported in the abstract. Only the corrected numbers make sense. Please adjust the numbers throughout the manuscript.

We agree and we have revised the manuscript accordingly (l. 342-344).

Finally, not even the same individuals/populations were used for both approaches (Table 1). That itself makes any sort of quantitative comparisons between the two methods questionable. Generally, I simply think that a well-designed RADseq study is just as valuable and will provide better data than a badly designed sequence-capture study.

We now refer to the ddRAD seq approach as a pilot study in the main text, and we explain the limits that led us to conclude that the approach was not promising.

Minor comments:

Title: Consider changing to “Towards population genomics in non-model species

with large genomes; a case study of the marine zooplankton *Calanus finmarchicus*”

We have changed the title accordingly.

p.3, ll.43: “The double digest RAD-seq, a modification of RAD-seq better suited for species with large genomes, ...”. This makes it sound as if ddRAD-seq was invented specifically for large genomes, while it really is generally more flexible and robust independent of genome size.

We have revised the text accordingly (l. 72-77).

Comments reviewer 2

Overall, Choquet et al. represents an interesting examination of two reduced representation methods for *Calanus* species in the Atlantic. Both the species and topic are extremely important and certainly worthy of publication. However I think the appeal of this work to a non-technical audience could be improved with some additional analyses and perhaps the movement of some of the methods to the supplement. Overall, I think this paper could be suitable for publication following revision, and re-review.

We agree and we have moved a large part of the method to the Supplementary Material (ddRAD-seq part + details of capture protocols). We have made new analyses from the SNPs data and focused more on the biological results of the study.

General Comments

1. Use of language is problematic particularly in the introduction. The work could benefit from a solid editing. Some of the issues have been noted but many have not.

We have revised the Introduction.

2. Overall the paper is largely technical in nature, and perhaps of limited appeal to a non-technical audience. It would have helped if some population genetic results and inferences could be included using the populations analyzed. This would certainly go a long way to increasing the novelty of the contribution and general appeal. The scale of sampling seems sufficient to do something of this type.

We agree and we have included results from FST, PCAs, genetic diversity, test for selection, that we have discussed.

3. The methods take a substantial amount of room. Again as this is not a technical journal, I would suggest that much of this is moved to the supplement and referenced within the main document.

We have done so by moving the ddRAD-seq part to the Supplement, as well as details of library preparation of capture methods.

Abstract

Ln 14. "Key" is vague, perhaps be more concise here.

We replaced it by "ecologically important species" (l .26)

Ln 15. Genome reduction is also unclear. Maybe use "reduced representation"

We replaced it by "genome reduced-representation" here and in other places in the text as well. (l .25)

Ln 26. The conclusion that capture is more efficient is of course partially dependent on enzyme choice. Might want to be careful how this is presented.

We have revised the text accordingly.

Introduction

Page 3 Ln 10. "May not" reflect **Corrected**

Page 3 Ln 24. "...consist of" **Corrected**

Page 3 Ln 35. Sentence is awkwardly worded. Please rephrase
We have rephrased.

Page 4. Ln 14. Consists of **Corrected**

Page 4. Ln 15. Delete "the approach of" **We have deleted.**

Page 4. Ln 21. Rephrase. "besides, the method.." **We have rephrased.**

Page 4. Ln 10. Why not compare the two approaches? Information on relative cost, time would also be useful. Why was individual RAD not attempted? Was DNA quantity or quality limiting?

DNA quantity and quality are the most limiting factors with such a small organism as Calanus. This is the reason why we had to pool. As the RAD-seq is now moved to the Supplementary, the practical comparison was not made.

Results

Page 11. Ln 2. Some of the population results should be included here such FSTs

etc, same below for sequence capture. Also when possible species comparison could be made with overlapping loci.

We have now provided these results for the capture approach. Loci in common between species have been used for analyses as well.

Discussion

Have other studies attempted population genomics in other marine copepods? If so they should be cited and discussed here.

We have cited examples studies that have attempted population genomics in copepods and marine zooplankton using genome-reduced representation protocols.

Comments reviewer 3

This is an entirely methodological paper that is missing any kind of biological results. The only results presented are the raw number of SNPs obtained with each approach, without any further validation or application of these marker sets. I would have expected at least some practical application of the two SNP sets to illustrate their usefulness to tackle the questions introduced at the beginning, i.e. analysis of population structure and gene flow (STRUCTURE, PCA, SFS-based demographic modeling, etc.).

We have strongly toned down the methodological aspect of the paper by moving substantial parts to the Supplementary Material. We have added results of analyses performed with the SNPs dataset and we discussed them.

The chosen approach of combining ddRAD-seq with Pool-seq seems very strange for the stated goal of detecting population structure, as it necessitates identifying population units before pooling. This makes it impossible to delineate hidden population structure or use individual-based approaches for analyzing population structure and gene flow. It is not clear from the description in the paper if this choice of sequencing method was driven primarily by insufficient DNA quantity obtained from each individual or by economic considerations. Moreover, Pool-seq critically relies on a large number of individuals in a given pool for accurate allele frequency estimation. The authors only used 16 individuals per pool and aimed for a total coverage of 200x,

which represents a very modest sequencing effort. Due to the lack of a good reference genome for in silico digestion, they ended up with a much higher number of loci with completely insufficient coverage for accurate SNP calling. Not surprisingly, only a very small number of SNPs could be called after applying a coverage filter. To solve this issue, the authors could have sequenced their libraries to higher depth, which would have easily resulted in a much higher number of useable SNPs. Therefore, the results from the ddRAD-seq approach are far from conclusive regarding the usefulness of the approach in this specific setting.

We agree and we have raised the point that the RAD approach can be prohibitive for species such as *Calanus* with a too large genome without any prior knowledge of it, and for which too little DNA available prevents single individual RADseq.

Even though the authors state that they are not aiming at directly comparing the two approaches, they still build most of their discussion around such a comparison of methods. However, their conclusions regarding the superiority of a sequence capture-based approach are not convincing due to the issues with their ddRAD approach mentioned above. Given these shortcomings, I would recommend to completely remove the ddRAD-seq part and instead focus on providing an in-depth description of a sequence capture approach to obtain genomic data in non-model organisms with large genome sizes. For such a methodological paper, however, it is critical that detailed protocols are provided as supporting information and custom analysis scripts are made publicly available.

We have removed the ddRAD-seq part from the main text, and provided supporting information as Supplementary Materials.

Moreover, the authors should address the problem of non-random selection of SNPs by sequence capture in the discussion. Most SNPs are expected to be within or close to genes when capturing with transcriptome-based baits, which means that selection might influence downstream population genetic analyses. Furthermore, the authors should provide a proof of concept by applying their SNP set to some commonly used analysis methods.

We have done a test for loci under selection with BAYESCAN, and it revealed only 0.007% and 0.006% loci potentially under selection for *C. finmarchicus* and *C. glacialis* respectively.

Appendix B

Dear Editor,

We would like to thank you and the reviewers for the constructive comments, which we found very helpful and relevant. We have now revised our manuscript accordingly. We have addressed all the points raised by the reviewers in this new version. Below are our responses to the primary issues raised by reviewers, as well as detailed responses.

We would also like to request your permission for adding Dr. Leocadio Blanco-Bercial to the author list. After our recent communication by email regarding this matter, I collected the approvals of all the current authors for adding Dr. Leocadio Blanco-Bercial as an author (see document attached), including the approval of Dr. Leocadio Blanco-Bercial himself. Dr. Leocadio Blanco-Bercial has contributed substantially to fixing the revisions asked by the reviewers, but he also helped to rethink some aspects of the manuscript. He has been writing some parts and has largely participated to the bioinformatic analyses. He has been involved in revising the manuscript and has approved the last version. As his expertise really helped us to deal with the challenging requests of the reviewers, we (all authors of the manuscript) consider that he deserves to be more than just acknowledged, and thus should be listed as a co-author of this manuscript.

Thank you very much for your support and your understanding.

Sincerely,

Dr. Marvin Choquet

Responses to reviewers' comments

Reviewer: 1

Minor comments:

1.) ll.117-120: "The enzymes for the digestion were chosen after in silico digestion of the very small fraction of the genome sequenced so far (< 0.5 %), resulting in a very high number of fragments, requiring a costly sequencing effort in order to achieve sufficient coverage."

I think the authors want to make the point that the number of obtained fragments was much higher than anticipated from their in silico digest here, but I don't think this really becomes clear and I don't understand why this would be the case (when properly done and extrapolated from the 0.5%). The authors themselves state in their response letter that according to the simRAD manual this is apparently not expected. I think the body size/ minimum DNA requirement is a much stronger argument and I would advise the authors to focus on that and say that they decided to go with targeted sequencing mainly because of that or provide a conclusive explanation why exactly their ddRAD-seq design failed despite the in silico digest.

Here we would like to clarify: as we would normally expect a small portion of genome to be sufficient for in-silico digestion, it was not successful enough in our case, and we consider that this is probably due to the large (and duplicated) genome of *C. finmarchicus*. Indeed, correspondence between in-silico and actual digestion fragments were not evaluated for cases of large complex duplicated genomes. In our case, we believe that having too many fragments constitutes an important reason to move to an alternative (i.e. capture). We have clarified this in the text (l. 122-131).

2.) ll. 372-376: Comparing heterozygosity levels at biallelic SNPs and (likely) multiallelic msats is flawed. The highest possible het. level based on the former is 0.5, whereas msats with more than two alleles will almost always show higher het. levels. Please revise.

We agree, and we have now removed the comparison. We are now using nucleotide diversity index values for comparisons with other studies.

3.) Does Figure 2 really make an important point that has to be in the main article? This seems to me more like supplementary information.

We have moved the Fig. 2 to Supplementary Material 2 (Supp. Fig. 3).

4.) Can you please make the color code for the different pops in Fig. 4 more distinct? It's hard to distinguish the pops at the moment.

We have revised Figure 4 (now Fig. 5). There are now only 2 colors (1 per species) and shapes are used for different locations.

Typos:

- l. 67: appear **Corrected (l. 69)**
- l. 85: targeted sequences **Corrected (l. 86-87)**
- l. 92: Capture probes sets have also proofed **Corrected (l. 95)**
- l. 93: delete "on" in "reported on" **Corrected (l. 96)**
- l. 125: "obtain an informative" **Corrected (l. 137)**
- l. 331: "is critical, but" **This sentence has been modified (l. 381)**
- l. 384: "shows" **We replaced "PCA" by "PCAs" (plural) (l. 436)**
- l. 417: "at the Canadian east coast" **Corrected (l. 486-487)**
- l. 419: "...detected any hybrids in any of the 85 locations..., though." **We removed "though" (l. 488)**

Reviewer: 3

(Alexander Nater, Department of Biology, University of Konstanz, Germany)

Line 68-69: It's not clear what the relationship between RAD-seq and allele-specific expression is. I guess the authors refer to problems with RNA-seq based SNP detection methods, but this should be clarified.

We clarified this by adding: "without allele-specific expression bias in contrast to RNA-seq". (l. 70-71)

Line 93: "Capture enrichments approaches have ..."

Corrected (l. 94-95).

Line 114-126: This paragraph is a bit awkward to read and needs some polishing.

The paragraph has been re-written (l. 116-138).

Line 231: "and mapped reads were realigned ..."

Corrected (l. 242).

Lines 234-236: Not clear if the SNP calling was performed for all individuals for each species or for both species together.

SNP calling was performed for all individuals of both species together. We clarified

by adding the following sentence: “Variants were called for all individuals of both species together at once using the HaplotypeCaller”. (l. 252-253)

Line 248-252: It doesn't make sense to apply a HWE filter to the SNPs if there is potential population structure in the data set, especially if the SNPs are later supposed to be used to test for such population structure.

We removed the HWE filter for the dataset aimed to be used in population genetics analyses. (section Methods: 4.2)

Moreover, applying minor allele frequency filters to the data will strongly bias the estimates of individual heterozygosity and nucleotide diversity. Such variant-specific filters should be avoided if the goal is to obtain accurate estimates of genetic diversity. In such cases, suitable filters should be applicable to both variant and non-variant sites, such as individual read depth, missingness, etc.

We removed minor allele frequency filters in the dataset aimed to calculate genetic diversity estimates (individual heterozygosity and nucleotide diversity).

Regarding the Nucleotide Diversity: we re-called the variants using GATK and forced GATK to also output the non-variant sites. We then filtered all the sites for depth of coverage and missingness (in order to keep only the sites covered >5x for 80% of genotypes). (See section Methods 4.1).

Line 249-250: Expected heterozygosity is a population statistic and not applicable at the individual level.

We calculated Individual Expected Heterozygosity levels on a per-individual basis. As we only have a few individuals, calculating the heterozygosity on a per-site basis would be problematic as each site would be very limited in the possible values that can be obtained. Therefore, we calculated the expected heterozygosity averaged over all variant sites in each individual. And we reported the range per species. (l. 265-268)

Line 251-252: It doesn't make sense to calculate nucleotide diversity on such small windows if the main goal is to compare genome-wide diversity among populations and species. Such small windows will only contain a very low number of variant sites and the window-wise estimates will be extremely noisy. A better approach would be to calculate nucleotide diversity across the entire assembly length and then use a resampling procedure to obtain confidence intervals. Moreover, it's not clear from the description how the authors obtained the proper number of callable yet non-variant sites within each window. For accurate estimates of nucleotide diversity, it's important to distinguish between well covered (i.e. callable) non-variant sites and sites that are not sufficiently covered (i.e. non-callable sites with unknown variant status). This can

be achieved quite easily by emitting both confident variant and non-variant sites during variant calling with GATK.

We now have recalled the variants together with the non-variant sites, filtered for coverage and missingness, and kept the window approach. We agree with the limitation of using small windows, however due to high fragmentation of the current reference assembly, which is re-enforced by the target capture approach, we could not find a better alternative. We now reported the genomic mean for each population directly in the figure (see Fig. 2). And see text in methods: l. 252-264.

Lines 298-300: I guess these values are the mean individual heterozygosities at variant sites, not the proportion of heterozygous sites?

Yes. See corrected text lines: l. 341-344.

Lines 309-311: Provide p-values for the global Fst estimates.

We provided the distribution of global Fst for each species, more informative than p-value:

We have changed our approach of the Fst estimates. For each species, after filtering by 5x coverage in sites present in at least 80% of the individuals, we selected one random SNP from each contig, to avoid biasing our estimates to longer contigs (with likely more variants sites with linkage issues). We repeated the procedure 1000 times, randomly selecting the SNPs in each iteration, and calculated the FST for each iteration. We are showing the distribution of the values obtained (Fig. 6). Our FST pattern still shows much larger FST for *C. glacialis* compared to *C. finmarchicus*. Only two of the 1000 iterations gave FST < 0 (-0.00056 and -0.00016) for *C. finmarchicus*. We consider that the distribution of the iterations is more informative than trying to obtain a p-value for the average or the median of the distribution (although we calculated it: $p < 0.001$). See methods, lines 358-362 and Fig. 6.

Lines 311-313: How do you explain the outliers for *C. Finmarchicus* in Fig. 4B? Are there any apparent quality differences compared to the other individuals that could point to technical reasons for this?

We checked for the two outliers and could explain their position in the PCA: the corresponding individuals are both from the Lurefjord, and they have the lowest numbers of raw reads sequenced and lowest values of individual mean depth of coverage. Under-representation of these two libraries in the total sequencing pool may be responsible for their position in the PCA. Overall, population from Lurefjord has less raw reads per individual, and we think the difficult sampling conditions in this fjord may be responsible for some degradation of the material and DNA of the sampled individuals. (l. 441-448).

Line 313: "..., consistent with ..."

Removed from the text.

Lines 331-332: It's not clear why the occurrence of high gene flow would necessitate a genomic approach.

High gene flow may be associated to subtle patterns of genetic structure, typical of zooplankton (high dispersal species; very large population sizes). Such a subtle structure may not be detectable with only a few markers, thus requiring a genomic approach.

We developed more in the text (see lines 382-385), and we cite three papers (Waples 1998; Willing et al. 2012, Blanco-Bercial and Bucklin 2016).

Lines 369-370: Tone down this statement. The selected approach for detecting outliers with BayeScan using only exonic SNPs is not suitable to reliably assess neutrality. At best, you can detect loci under recent and strong positive selection. I would like to see a site frequency spectrum of all SNPs for the two species in the Supplementary Materials to get a better impression of potential selective constraints affecting these SNPs.

We toned down the statement, see l. 424-425.

We did the site frequency spectrum of all SNPs for the 2 species, see in Supp Mat 2 (Supp Fig. 4 & 5). In both species, the profiles did not appear to show signs of selection, but more a somewhat flat profile. The low number of individuals makes however difficult a precise interpretation, especially for *C. glacialis*.

Lines 371-373: It's not clear how these heterozygosities have been calculated. Were they averaged over all positions in the two data sets that were polymorphic relative to the reference assembly? If so, there is an obvious downwards bias for *C. glacialis*, as this species would probably have quite a few fixed differences to the *C. finmarchicus* reference. These sites would have a mean individual heterozygosity of 0, as they are not polymorphic in *C. glacialis*. Anyways, SNP heterozygosity is not a suitable statistic to compare levels of genetic diversity between populations or species and should not be compared to heterozygosities for highly variable markers such as microsatellites. The proper statistic to assess genetic diversity is nucleotide diversity.

We have acknowledged the potential bias for *C. glacialis*. (lines 454-462). And we have now used values of Nucleotide Diversity to compare genetic diversity with results from other studies. (l. 427-432)

Lines 394-395: A larger number of markers will lead to higher precision, not necessarily

higher accuracy.

Corrected - I. 466.

Line 403-404: Again, SNP heterozygosity is not a proper statistic to measure genetic diversity. If you want to compare genetic diversity across studies and species, nucleotide diversity should be used. For example, a population expansion in *Centropages typicus* could easily explain the lower levels of SNP heterozygosity despite similar or higher levels of genetic diversity.

We have removed the comparison using SNP heterozygosity.

Table 2: Also provide mean sequencing depth for each individual after filtering.

We have provided the individual values of depth of coverage averaged over the list of contigs on target (after filtering). See Table 2.

Fig. 3: “Individual heterozygosity” instead of “Expected heterozygosity”.

Revised.

Appendix C

Dear Editor,

We would like to thank you and the reviewer for the constructive comments, which we found very helpful and relevant. We have now revised our manuscript accordingly. We have addressed all the points raised by the reviewer 3 in this new version. Below are our detailed responses to the issues raised by the reviewer 3.

Sincerely,

Dr. Marvin Choquet

Responses to the reviewer's comments

Reviewer: 3

Line 255: "... to also output ... in the outputted VCF file.". I would suggest replacing "outputted" by "resulting".

Corrected (l. 255).

Line 256-257: "we filtered the sites to keep only those covered 5x or more and present in at least 80% of the individuals". The current description sounds as if total coverage per site needed to be $\geq 5x$, rather than coverage per individual. Please clarify what is meant by a site being present in at least 80% of individuals.

We clarified as follow:

"Using VCFtools (version 0.1.13) [53], we filtered the sites to keep only those with mean depth values (over all individuals) greater than or equal to 5x. Among these, sites with more than 20% of missing data were excluded, which means that we kept only the sites represented in at least 80% of the genotypes." (l. 255-259).

Line 260-264: Omit "We considered that the number of individuals was too low to make accurate estimates of nucleotide diversity on a per-site basis, as each site is very limited in the possible values that can be obtained. To get around this,". It is obvious that you do not want to report nucleotide diversity for each site separately. Nonetheless, it is important to clarify that nucleotide diversity was estimated on a per-site basis and averaged in 780-bp windows using only the sites that passed the filtering. The description "we measured diversity in non-overlapping windows of 780 bp length" is confusing here.

We have removed the first part and replaced the rest by: "Nucleotide diversity was estimated on a per-site basis and averaged in 780-bp windows (average of contig size distribution) using only the sites that passed the filtering. We reported the mean of π across windows for each population, with VCFtools (version 0.1.13)." (l. 261-264).

Line 264: Mean of π across windows for each population?

Corrected (l. 263)

Lines 265-268: Same issue as above. The calculation would still be on a per-SNP level, but you only report genome-wide averages per individual.

We have re-written the sentence: “Observed heterozygosities (proportion of heterozygous sites) at variant sites were calculated on a per-SNP level in each individual and averaged over all positions present in both species together, using VCFtools (option --het; version 0.1.13).”

(l. 265-267)

Line 266-267: Reporting average observed heterozygosities per individual is fine. My point was referring to the term “individual expected heterozygosity”. Expected heterozygosity is a population-level statistic, it simply cannot be calculated on a per-individual basis. What you are calculating here is observed heterozygosity, i.e. you are calculating the proportion of heterozygous SNP for each individual. Expected heterozygosity would be the expected proportion of heterozygous genotypes given the estimated allele frequencies in a population.

We have re-written this part and clarified: “Observed heterozygosities (proportion of heterozygous sites) at variant sites were calculated on a per-SNP level in each individual and averaged over all positions present in both species together, using VCFtools (option --het; version 0.1.13).” (l. 265-267).

Line 276: “variants phased” instead of “haplotypes phased”

Corrected. (l. 275)

Line 278: “Variants present in less than 80% of genotypes were filtered out.” It is not clear what this is supposed to mean. It reads as if SNPs with less than 80% of genotypes carrying the alternative allele were filtered out, but that would not make much sense at all.

We clarified by using “Sites” instead of “Variants”. (l. 276)

Line 292: Custom scripts should either be provided in the Supplementary Materials or uploaded to a suitable repository.

The mentioned PERL script is available in a repository online, and a reference has been added in the text. (l. 291)

The complementary script used for calculating FST has been added to the Supplementary Material 3. (l. 295)

Lines 427-434: Comparing absolute values of nucleotide diversity between autosomal SNPs and mitochondrial genes is quite pointless, as nucleotide diversity depends on the mutation rate, which is obviously different between these marker types.

This part has now been removed from the discussion.

Line 451: "(PC1: ~12% of variance explained)."

Corrected. (l. 441).